# Efficient Combinatorial Optimization via Heat Diffusion

**Hengyuan Ma**

Institute of Science and Technology for Brain-inspired Intelligence
Fudan University
Shanghai, China 200433
`hangyuanma21@m.fudan.edu.cn`

**Wenlian Lu**

Institute of Science and Technology for Brain-inspired Intelligence
Fudan University
Shanghai, China 200433
`wenlian@fudan.edu.cn`

**Jianfeng Feng**

Institute of Science and Technology for Brain-inspired Intelligence
Fudan University
Shanghai, China 200433
`jianfeng64@gmail.com`

## Abstract

Combinatorial optimization problems are widespread but inherently challenging due to their discrete nature. The primary limitation of existing methods is that they can only access a small fraction of the solution space at each iteration, resulting in limited efficiency for searching the global optimal. To overcome this challenge, diverging from conventional efforts of expanding the solver's search scope, we focus on enabling information to actively propagate to the solver through heat diffusion. By transforming the target function while preserving its optima, heat diffusion facilitates information flow from distant regions to the solver, providing more efficient navigation. Utilizing heat diffusion, we propose a framework for solving general combinatorial optimization problems. The proposed methodology demonstrates superior performance across a range of the most challenging and widely encountered combinatorial optimizations. Echoing recent advancements in harnessing thermodynamics for generative artificial intelligence, our study further reveals its significant potential in advancing combinatorial optimization. The codebase of our study is available in `https://github.com/AwakerMhy/HeO`.

## 1 Introduction

Combinatorial optimization problems are prevalent in various applications, encompassing circuit design [1], machine learning [2], computer vision [3], molecular dynamics simulation [4], traffic flow optimization [5], and financial risk analysis [6]. This widespread application creates a significant demand for accelerated solutions to these problems. Alongside classical algorithms, which encompass both exact solvers and metaheuristics [7], recent years have seen remarkable advancements in addressing combinatorial optimization. These include quantum adiabatic approaches [8, 9, 10], simulated bifurcation [11, 12, 13], coherent Ising machine [14, 15], high-order Ising machine [16],

and deep learning techniques [17, 18]. However, due to the exponential growth of the solution number, finding the optima within a limited computational budget remains a daunting challenge.

Our primary focus is on iterative approximation solvers, which constitute a significant class of combinatorial optimization methods. An iterative approximation solvers typically begin with an initial solution and iteratively improve it by finding better solutions within the neighborhood of the current solution, known as the search scope or more vividly, *receptive field*. However, due to combinatorial explosion, as the scope of the receptive field increases, the number of solutions to be assessed grows exponentially, making a thorough evaluation of all these solutions expensive. As a result, current approaches are limited to a narrow receptive field, rendering them blind to distant regions in the solution space and heightening the risk of getting trapped in local minimas or areas with bumpy landscapes. Although methods like large neighborhood search [19], variable neighborhood search [20] and path auxiliary sampling [21] are designed to broaden the search scope, they can only gather a modest increment of information from the expanded search scope. Consequently, the current solvers' receptive field remains significantly constrained, impeding their efficiency.

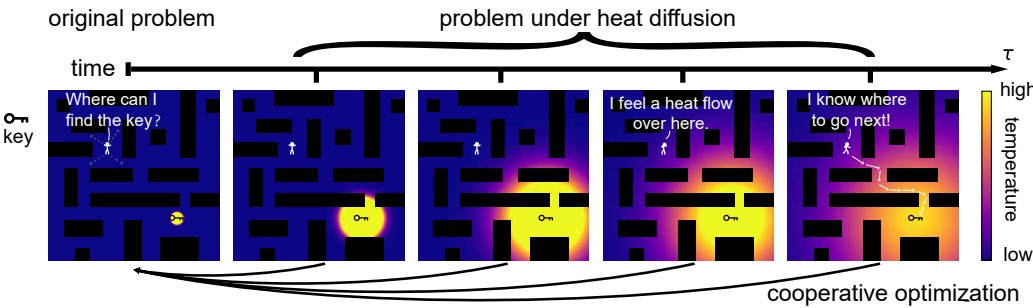

Figure 1: **The heat diffusion optimization (HeO) framework.** The efficiency of searching a key in a dark room is significantly improved by employing navigation that utilizes heat emission from the key. In our framework, heat diffusion transforms the target function of a combinatorial optimization problem into different versions while preserving the location of the optima. Therefore, the gradient information of these transformed functions cooperatively help to optimize the original target function.

In this study, we approach the prevalent limitation stated above from a unique perspective. Instead of expanding the solver's receptive field to acquire more information from the solution space, we concentrate on propagating information from distant areas of the solution space to the solver via heat diffusion [22]. To illustrate, imagine a solver searching for the optima in the solution space akin to a person searching for a key in a dark room, as depicted in Fig. 1. Without light, the person is compelled to rely solely on touching his surrounding space. The tactile provides only localized information, leading to inefficient navigation. This mirrors the current situation in combinatorial optimization, wherein the receptive field is predominantly confined to local information. However, if the key were to emit heat, its radiating warmth would be perceptible from a distance, acting as a directional beacon. This would significantly enhance navigational efficiency for finding the key.

Motivated by the above metaphor, we propose a simple but efficient framework utilizing heat diffusion to solve various combinatorial optimization problems. Heat diffusion transforms the target function into different versions, within which the information from distant regions actively flow toward the solver. Crucially, the backward uniqueness of the heat equation [23] guarantees that the original problem's optima are unchanged under these transformations. Therefore, information of target functions under heat diffusion transformations of different degrees can be cooperatively employed for optimize the original problem (Fig. 1). Empirically, our framework demonstrates superior performance compared to advanced algorithms across a diverse range of combinatorial optimization instances, spanning quadratic to polynomial, binary to ternary, unconstrained to constrained, and discrete to mixed-variable scenarios. Mirroring the recent breakthroughs in generative artificial intelligence through diffusion processes [24], our research further reveals the potential of heat diffusion, a related thermodynamic phenomenon, in enhancing combinatorial optimization.

## 2  Failure of gradient-based combinatorial optimization

We will first formulate the general combinatorial optimization problem and then reframe it in a gradient descent-based manner. This reformulation allows us to utilize heat diffusion later. Various combinatorial optimization problems can be naturally formalized as a pseudo-Boolean optimization (PBO) problem [25], in which we aim to find the minima of a real-value target function $f \in \mathbb{R}^n \mapsto \mathbb{R}$ subjecting to a binary constraints

$$\min_{\mathbf{s} \in \{-1,1\}^n} f(\mathbf{s}), \tag{1}$$

where $\mathbf{s}$ is binary configuration (bits), and $f(\cdot)$ is the target function. Through the transformation $(\mathbf{s}+1)/2$, our definition aligns with that in [26], where elements of $\mathbf{s}$ take 0 or 1. Given the advanced development of gradient-based algorithms, we are interested in converting the discrete optimization problem into a differentiable one, thereby enabling gradient descent. To achieve this purpose, we encode the bits $s_i$, $i = 1, \ldots, n$ as independent Bernoulli variables $p(s_i = \pm 1|\boldsymbol{\theta}) = 0.5 \pm (\theta_i - 0.5)$ with $\theta_i \in [0, 1]$. In this way, we convert the original combinatorial optimization problem into

$$\min_{\boldsymbol{\theta} \in \mathcal{I}} h(\boldsymbol{\theta}), \tag{2}$$

where $\mathcal{I} := [0, 1]^n$, and $h(\boldsymbol{\theta}) = \mathbb{E}_{p(\mathbf{s}|\boldsymbol{\theta})}[f(\mathbf{s})]$. The minima $\boldsymbol{\theta}^*$ of Eq. (2) is $\boldsymbol{\theta}^* = 0.5(\mathrm{sgn}(\mathbf{s}^*) + 1)$, given that $\mathbf{s}^*$ is a minima of the original problem Eq. (1). Here, $\mathrm{sgn}(\cdot)$ is the element-wise sign function. Now Eq. (2) can be solved through gradient descent starting from a given initial $\boldsymbol{\theta}_0$

$$\boldsymbol{\theta}_{t+1} = \boldsymbol{\theta}_t - \gamma \bigtriangledown_{\boldsymbol{\theta}} h(\boldsymbol{\theta}_t), \quad t = 1, \ldots, T, \tag{3}$$

where $\gamma$ is the learning rate and $T$ is the iteration number. Unfortunately, this yields a probability distribution $p(\mathbf{s}|\boldsymbol{\theta})$ over the configuration space $\{-1, 1\}^n$, instead of a deterministic binary configuration $\mathbf{s}$ as desired. Although we can manually binarize $\boldsymbol{\theta}$ through $\mathbf{B}(\boldsymbol{\theta}) := \mathrm{sgn}(\boldsymbol{\theta} - 0.5)$ to get the binary configuration which maximizes probability $p(\mathbf{s}|\boldsymbol{\theta})$, the outcome $f(\mathbf{B}(\boldsymbol{\theta}))$ may be much higher than $h(\boldsymbol{\theta})$, resulting in significant performance degradation [27]. This suggests that a good gradient-based optimizer should efficiently diminish the uncertainty in the output distribution $p(\mathbf{s}|\boldsymbol{\theta})$, which can be measured by its total variance

$$V(\boldsymbol{\theta}) = \sum_{i=1}^{n} \theta_i (1 - \theta_i). \tag{4}$$

### 2.1  Monte Carlo gradient estimation

Conventionally, we can solve the problem Eq. (2) by approximating the gradient of $h(\boldsymbol{\theta})$ via Monte Carlo gradient estimation (MCGE) [28] (Alg. 2, *Appendix*), in which we estimate the gradient in Eq. (3) as

$$\bigtriangledown_{\boldsymbol{\theta}} h(\boldsymbol{\theta}) = \mathbb{E}_{p(\mathbf{s}|\boldsymbol{\theta})}[f(\mathbf{s}) \bigtriangledown_{\boldsymbol{\theta}} \log p(\mathbf{s}|\boldsymbol{\theta})] \approx \frac{1}{M} \sum_{m=1}^{M} f(\mathbf{s}^{(m)}) \bigtriangledown_{\boldsymbol{\theta}} \log p(\mathbf{s}^{(m)}|\boldsymbol{\theta}), \tag{5}$$

where $\mathbf{s}^{(m)} \sim_{i.i.d.} p(\mathbf{s}|\boldsymbol{\theta})$, $m = 1, \ldots, M$. However, it turns out that MCGE performs poorly even equipped with momentum, compared to existing solvers such as simulated annealing and Hopfield neural network, as shown in Fig. 2. We interpret this result as follows. Although MCGE turns the combinatorial optimization problem into a differentiable one, it does not reduce any inherent complexity of the original problem, which may contains a convoluted landscape. As gradient only provides local information, MCGE is susceptible to be trapped in local minimas.

## 3  Heat diffusion optimization

The inferior performance of the MCGE is attributed to the narrow receptive field of the gradient descent. To overcome this drawback, we manage to provide more efficient navigation to the solver by employing heat diffusion [29], which propagates information from distant region to the solver. Intuitively, consider the parameter space as a thermodynamic system, where each parameter $\boldsymbol{\theta}$ is referred to as a location and is associated with an initial temperature value $-h(\boldsymbol{\theta})$, as shown in Fig. 1.

Then the optimization procedure can be described as the process that the solver is walking around the parameter space to find the location $\boldsymbol{\theta}^*$ with the highest temperature (or equivalently, the global minima of $h(\boldsymbol{\theta})$). As time progresses, heat flows obeying the Newton's law of cooling, leading to an evolution of the temperature distribution across time. The heat at $\boldsymbol{\theta}^*$ flows towards surrounding areas, ultimately reaching the solver's location. This provides valuable information for the solver, as it can trace the direction of heat flow to locate the $\boldsymbol{\theta}^*$.

### 3.1 Heat diffusion on the parameter space

Now we introduce heat diffusion for combinatorial optimization. We extent the parameter space of $\boldsymbol{\theta}$ from $[0,1]^n$ to $\bar{\mathbb{R}}^n$ with $\bar{\mathbb{R}} = \mathbb{R} \cup \{-\infty, +\infty\}$. To keep the probabilistic distribution $p(\mathbf{s}|\boldsymbol{\theta})$ meaningful for $\boldsymbol{\theta} \notin [0,1]^n$, we now redefine $p(s_i = \pm 1|\boldsymbol{\theta}) = \mathrm{clamp}(0.5 \pm (\theta_i - 0.5), 0, 1)$, where the clamp function is calculated as $\mathrm{clamp}(x, 0, 1) = \max(0, \min(x, 1))$. Denote the temperature at location $\boldsymbol{\theta}$ and time $\tau$ as $u(\tau, \boldsymbol{\theta})$, which is the solution to the following unbounded heat equation [29]

$$\begin{cases} \partial_\tau u(\tau, \boldsymbol{\theta}) & = & \Delta_{\boldsymbol{\theta}} u(\tau, \boldsymbol{\theta}), & \tau > 0, & \boldsymbol{\theta} \in \mathbb{R}^n \\ u(\tau, \boldsymbol{\theta}) & = & h(\boldsymbol{\theta}), & \tau = 0, & \boldsymbol{\theta} \in \mathbb{R}^n \end{cases}, \tag{6}$$

where $\Delta$ is the Laplacian operator: $\Delta g(\mathbf{x}) = \sum_{i=1}^n \partial_{x_i} g(\mathbf{x})$. For $\boldsymbol{\theta} \in \bar{\mathbb{R}}^n/\mathbb{R}^n$, we define $u(\tau, \boldsymbol{\theta}) = \lim_{\boldsymbol{\theta}_n \to \boldsymbol{\theta}} u(\tau, \boldsymbol{\theta}_n)$, where $\{\boldsymbol{\theta}_n\}$ is a sequence in $\mathbb{R}^n$ converged to $\boldsymbol{\theta}$. Heat equation in the combinatorial optimization exhibits two beneficial characteristics. Firstly, the propagation speed of heat is infinite [22], implying that the information can reach the solver instantaneously. Secondly, the location of the global minima does not change across time $\tau$, as demonstrated in the following theorem.

**Theorem 1.** *For any $\tau > 0$, the function $u(\tau, \boldsymbol{\theta})$ and $h(\boldsymbol{\theta})$ has the same global minima in $\bar{\mathbb{R}}^n$*

$$\arg\min_{\boldsymbol{\theta} \in \bar{\mathbb{R}}^n} u(\tau, \boldsymbol{\theta}) = \arg\min_{\boldsymbol{\theta} \in \bar{\mathbb{R}}^n} h(\boldsymbol{\theta}) \tag{7}$$

Consequentially, we can generalize the gradient descent approach Eq. (3) by substituting the function $h(\boldsymbol{\theta})$ with $u(\tau_t, \boldsymbol{\theta})$ for different $\tau_t > 0$ at different iteration step $t$ in Eq. (3), as follows

$$\boldsymbol{\theta}_{t+1} = \boldsymbol{\theta}_t - \gamma \bigtriangledown_{\boldsymbol{\theta}} u(\tau_t, \boldsymbol{\theta}_t), \tag{8}$$

where the subscript '$_t$' in $\tau_t$ means that $\tau_t$ can vary across different steps. In this way, the solver can receive the gradient information about distant region of the landscape that is propagated by the heat diffusion, resulting in a more efficient navigation. However, Eq. (8) will converge to $\check{\theta}_i = \begin{cases} +\infty, & s_i^* = +1 \\ -\infty, & s_i^* = -1 \end{cases}$ , since $\boldsymbol{\theta}_t$ are unbounded. To make the procedure Eq. (8) practicable, we project the $\boldsymbol{\theta}_t$ back to $\mathcal{I}$ after gradient descent at each iteration

$$\boldsymbol{\theta}_{t+1} = \mathrm{Proj}_{\mathcal{I}}\big(\boldsymbol{\theta}_t - \gamma \bigtriangledown_{\boldsymbol{\theta}} u(\tau_t, \boldsymbol{\theta}_t)\big), \tag{9}$$

so that $\boldsymbol{\theta}_t \in \mathcal{I}$ always holds, where we define the projection as $\mathrm{Proj}_{\mathcal{I}}(\mathbf{x})_i = \min(1, \max(0, x_i))$ for $i = 1, \dots, n$ and for $\mathbf{x} \in \mathbb{R}^n$. Eq. (9) is a reasonable update rule for finding the minimum $\boldsymbol{\theta}^*$ within $\mathcal{I}$ for the following two reasons: (1) The projection of the $\check{\boldsymbol{\theta}}$ is the minimum of $h(\boldsymbol{\theta})$ in $\mathcal{I}$, i.e., $\mathrm{Proj}_{\mathcal{I}}(\check{\boldsymbol{\theta}}) = \boldsymbol{\theta}^*$; (2) Due to the property of the projection, if the solver moves towards $\check{\boldsymbol{\theta}}$, it also gets closer to $\boldsymbol{\theta}^*$. More importantly, since the coordinates of $\check{\boldsymbol{\theta}}$ are all infinite, the convergent point of Eq. (9) must be one of the vertices of $\mathcal{I}$, i.e., $\{0, 1\}^n$. This suggests that Eq. (9) tends to give an output $\boldsymbol{\theta}$ that diminishes the uncertainty $V(\boldsymbol{\theta})$ (Eq. (4)).

### 3.2 Solving the heat equation

To develop an algorithm for solving the combinatorial optimization problem from Eq. (9), we must solve the heat equation Eq. (6), which seems a significant challenge when the dimension $n$ is high. Fortunately, the solution has a closed form if the target function $f(\mathbf{s})$ can be written as a multi-linear polynomial of $\mathbf{s}$

$$f(\mathbf{s}) = a_0 + \sum_{i_1} a_{1,i_1} s_{i_1} + \sum_{i_1 < i_2} a_{2,i_1 i_2} s_{i_1} s_{i_2} + \cdots + \sum_{i_1 < \cdots < i_K} a_{K,i_1 \dots i_K} s_{i_1} \cdots s_{i_K}, \tag{10}$$

a condition met by a wide range of combinatorial optimization problems [16].

**Theorem 2.** *Supposed that $f(\mathbf{s})$ is a multilinear polynomial of $\mathbf{s}$, then the solution to Eq. (6) is*

$$u(\tau, \boldsymbol{\theta}) = \mathbb{E}_{p(\mathbf{x})}[f(\mathrm{erf}(\frac{\boldsymbol{\theta} - \mathbf{x}}{\sqrt{\tau}}))], \quad \mathbf{x} \in \mathrm{Unif}[0, 1]^n, \tag{11}$$

*where $\mathrm{erf}(x) = \frac{2}{\sqrt{\pi}} \int_0^x e^{-t^2} dt$ is the error function.*

For more general cases other then Eq. (10) (such as Eq. (13)), we still use the approximation (Eq. (11)).

### 3.3 Proposed algorithm

Based on Eq. (9) and Thm. 2, we proposed *heat diffusion optimization (HeO)*, a gradient-based algorithm for combinatorial optimization, as illustrated in Alg. 1, where we estimate Eq. (11) with one sample $\mathbf{x}_t \sim \mathrm{Unif}[0, 1]^n$, and we denote $\sqrt{\tau}_t = \sigma_t$ for short. Our HeO can be equipped with momentum, which is shown in Alg. 3 in *Appendix*. In contrast to those methods designed for solving special class of PBO (Eq. (1)) such as quadratic unconstrained binary optimization (QUBO), our HeO can directly solve PBO problems with general form. Although PBO can be represented as QUBO [30], this necessitates the introduction of auxiliary variables, which may consequently increase the problem size and leading to additional computational overhead [31]. Compared to other algorithms, our HeO has relatively low complexity. The most computationally intensive operation at each step is gradient calculation, which can be explicitly expressed or efficiently computed with tools like PyTorch's autograd, and can be accelerated using GPUs. As shown in Fig. S1 of the in *Appendix*, the time cost per iteration of our methods increases linearly with the problem dimension, with a small constant coefficient. Therefore, our HeO is efficient even in high-dimensional cases.

---

**Algorithm 1** Heat diffusion optimization (HeO)

---

**Input:** target function $f(\cdot)$, step size $\gamma$, $\sigma$ schedule $\{\sigma_t\}$, iteration number $T$
initialize elements of $\boldsymbol{\theta}_0$ as 0.5
**for** $t = 0$ **to** $T - 1$ **do**
    sample $\mathbf{x}_t \sim \mathrm{Unif}[0, 1]^n$
    $\mathbf{g}_t \leftarrow \nabla_{\boldsymbol{\theta}} f(\mathrm{erf}(\frac{\boldsymbol{\theta}_t - \mathbf{x}_t}{\sigma_t}))$
    $\boldsymbol{\theta}_{t+1} \leftarrow \mathrm{Proj}_{\mathcal{I}}(\boldsymbol{\theta}_t - \gamma \mathbf{g}_t)$
**end for**
**Output:** binary configuration $\mathbf{s}_T = \mathrm{sgn}(\boldsymbol{\theta}_T - 0.5)$

---

One counter-intuitive thing is that to minimize the target function $h(\boldsymbol{\theta})$, the HeO actually are minimizing different functions $u(\tau_t, \boldsymbol{\theta})$ at different step $t$. We interpret this by providing an upper bound for the target optimization loss $h(\boldsymbol{\theta}) - h(\boldsymbol{\theta}^*)$.

**Theorem 3.** *Denote $\check{f} = \max_{\mathbf{s}} f(\mathbf{s})$. Given $\tau_2 > 0$ and $\epsilon > 0$, there exists $\tau_1 \in (0, \tau_2)$, such that*

$$h(\boldsymbol{\theta}) - h(\boldsymbol{\theta}^*) \leq \left[ (\check{f} - f^*) (u(\tau_2, \boldsymbol{\theta}) - u(\tau_2, \boldsymbol{\theta}^*) + \frac{n}{2} \int_{\tau_1}^{\tau_2} \frac{u(\tau, \boldsymbol{\theta}) - u(\tau, \boldsymbol{\theta}^*)}{\tau} d\tau) \right]^{1/2} + \epsilon. \tag{12}$$

Accordingly, minimizing $u(\tau, \boldsymbol{\theta})$ for each $\tau$ cooperatively aids in minimizing the original target function $h(\boldsymbol{\theta})$. Thus, we refer to HeO as a *cooperative optimization* paradigm, as illustrated in Fig. 1.

## 4 Experiments

We apply our HeO to a variety of NP-hard combinatorial optimization problems to demonstrate its broad applicability. Unless explicitly stated otherwise, we employ the $\tau_t$ schedule as $\sqrt{\tau_t} =: \sigma_t = \sqrt{2}(1 - t/T)$ for HeO throughout this work. The sensitivity of other parameter settings including the step size $\gamma$ and iterations $T$ are shown in Fig. S2. This choice is motivated by the idea that the reversed direction of heat flow guides the solver towards the original of its source, i.e., the global minima. Noticed that this choice is not theoretically necessary, as elaborated in *Discussion*.

**Toy example.** We consider the following target function

$$f(\mathbf{s}) = \mathbf{a}_2^{\mathrm{T}} \mathrm{sigmoid}(W\mathbf{s} + \mathbf{a}_1) \tag{13}$$

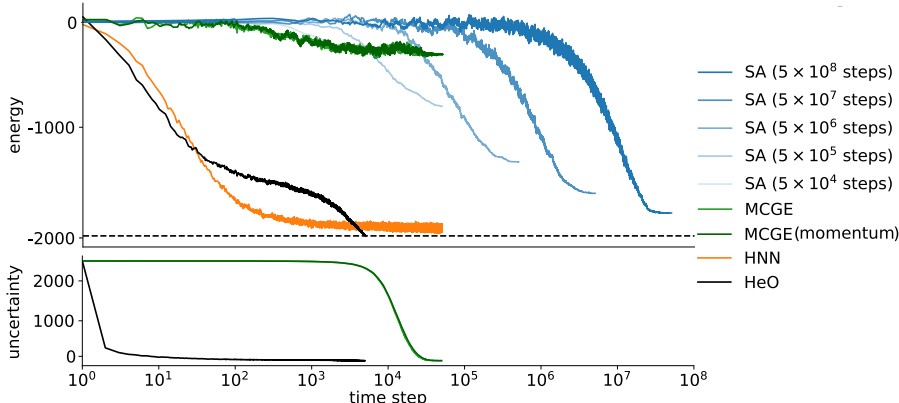

Figure 2: Performance of HeO (Alg. 3, *Appendix*), Monte Carlo gradient estimation (MCGE), Hopfield neural network (HNN) and simulated annealing (SA) on minimizing the output of a neural network (Eq. (13)). Top panel: the target function. Bottom panel: the uncertainty $V(\boldsymbol{\theta})$ (Eq. (4)).

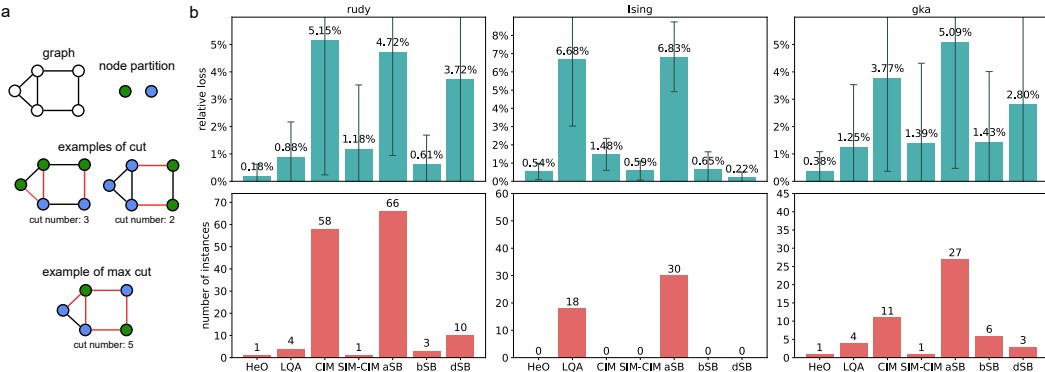

Figure 3: a, Illustration of the max-cut problem. b, Performance of HeO (Alg. 1) and representative iterative approximation methods including LQA [10], aSB [12], bSB [13], dSB [13], CIM [35] and SIM-CIM [15] on max-cut problems from the Biq Mac Library [36]. Top panel: average relative loss for each algorithm over all problems. Bottom panel: the count of instances where each algorithm ended up with one of the bottom-2 worst results among the 7 algorithms.

where $\text{sigmoid}(x) = \frac{1}{1+e^{-x}}$, and the elements of the network parameters $\mathbf{a}_1 \in \mathbb{R}^n$, $\mathbf{a}_2 \in \mathbb{R}^m$, $W \in \mathbb{R}^{m,n}$ are uniformly sampled from $[-1, 1]$ and fixed during optimizing. According to the universal approximation theory [32], $f(\mathbf{s})$ can approximate any continuous function with sufficiently large $m$, thereby representing a general target function. We compare the performance of our HeO with momentum (Alg. 3) against several representative methods: the conventional gradient-based solver MCGE [28] (with or without momentum), the simulated annealing [33], and the Hopfield neural network [34]. As shown in Fig. 2, our HeO demonstrates exceptional superiority over all other methods, and efficiently reduces its uncertainty compared to MCGE.

**Quadratic unconstrained binary optimization (QUBO).** QUBO is the combinatorial optimization problem with quadratic target function ($J \in \mathbb{R}^{n \times n}$ is a symmetric matrix with zero diagonals)

$$f(\mathbf{s}) = \mathbf{s}^{\mathrm{T}} J \mathbf{s}. \tag{14}$$

This corresponds to the case where $K = 2$ in Eq. (10). A well-known class of QUBO is max-cut problem [27], in which we divide the vertices of a graph into two distinct subsets and aim to maximize the number of edges between them. Its target function is expressed as Eq. (14), where $J$ is determined by the adjacency matrix of the graph.

We compare our HeO with representative iterative approximation methods especially developed for solving QUBO including LQA [10], aSB [12], bSB [13], dSB [13], CIM [35], and SIM-CIM [15]

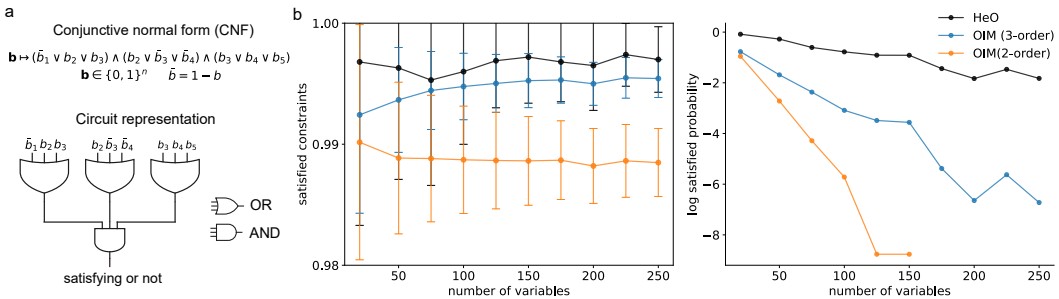

Figure 4: a, Illustration of the boolean 3-satisfiability (3-SAT) problem. b, Performance of HeO (Alg. 4, *Appendix*), 2-order and 3-order oscillation Ising machine (OIM) [16] on 3-SAT problems with various number of variables from the SATLIB [37]. We report the mean percent of constraints satisfied (left) and probability of satisfying all claims (right) for each algorithm.

on max-cut problems in the Biq Mac Library [36][1]. We report the relative loss averaged over all instances and the count of the instances where each algorithm gives the bottom-2 worse result among the 7 algorithms. As shown in Fig. 3, our HeO is superior to other methods in terms of two metrics.

**Polynomial unconstrained binary optimization (PUBO).** PUBO is a class of combinatorial optimization problems, in which higher-order interactions between bits $s_i$ appears in the target function. Existing methods for solving PUBO fall into two categories: the first approach involves transforming PUBO into QUBO by adding auxiliary variables through a quadratization process, and then solving it as a QUBO problem [38], and the one directly solves PUBO [16]. Quadratization may dramatically increases the dimension of the problem, hence brings heavier computational overhead, while our HeO can be directly used for solving PUBO. A well-known class of PUBO is the Boolean 3-satisfiability (3-SAT) problem [27], which involves determining the satisfiability of a Boolean formula over $n$ Boolean variables $b_1, \ldots, b_n$ where $b_i \in \{0, 1\}$. The Boolean formula is structured in Conjunctive Normal Form (CNF) consisting of $H$ conjunction ($\wedge$) of clauses, and each clause $h$ is a disjunction ($\vee$) of exactly three literals (either a Boolean variable or its negation). An algorithm of 3-SAT aims to find the Boolean variables that makes as many as clauses satisfied.

To apply our HeO to the 3-SAT, we encode each Boolean variable $b_i$ as $s_i$, which is assigned with value 1 if $b_i = 1$, otherwise $s_i = -1$. For a literal, we define a value $c_{h_i}$, which is $-1$ if the literal is the negation of the corresponding Boolean variable, otherwise it is 1. Then finding the Boolean variables that satisfies as many as clauses is equivalent to minimize the target function

$$f(\mathbf{s}) = \sum_{h=1}^{H} \prod_{i=1}^{3} \frac{1 - c_{h_i} s_{h_i}}{2}. \tag{15}$$

This corresponds to the case where $K = 3$ in Eq. (10). We compared our HeO (Alg. 4, *Appendix*) with the second-order oscillator Ising machines (OIM) solver that using quadratization and the state-of-art 3-order OIM proposed in [16] on 3-SAT instance in SATLIB [2]. As shown in Fig. 4, our HeO is superior to other methods in attaining higher quality of solutions and finding more the complete satisfiable solution (solutions that satisfying all clauses). Notably, for the cases of 175-250 variables, our HeO is able to find more complete satisfiable solutions, compared to the 3-order OIM, while the 2-order OIM fails to find any complete solutions [16].

**Ternary optimization.** Neural networks excel in learning and modeling complex functions, but they also bring about considerable computational demands due to their vast number of parameters. A promising strategy to mitigate this issue is quantization, which converts network parameters into discrete values [39]. However, directly training networks with discrete parameters introduces a significant challenge due to the high-dimensional combinatorial optimization problem it presents.

We apply our HeO and MCGE to directly train neural networks with ternary value $(-1, 0, 1)$. Supposed that we have an input-output dataset $\mathcal{D}$ generated by a ground-truth ternary single-layer per-

---

[1] https://biqmac.aau.at/biqmaclib.html
[2] https://www.cs.ubc.ca/~hoos/SATLIB/benchm.html

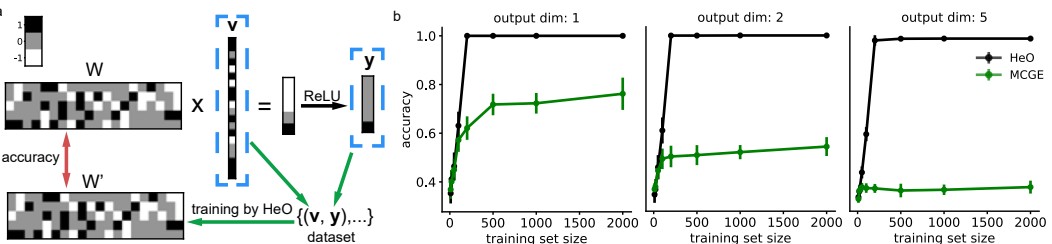

Figure 5: a, Training networks with ternary-value parameters. b, The weight value accuracy of the HeO (Alg. 5, *Appendix*) and Monte Carlo gradient estimation (MCGE) with momentum under different sizes of training set ($n = 100, m = 1, 2, 5$). We estimate the mean and std from 10 runs.

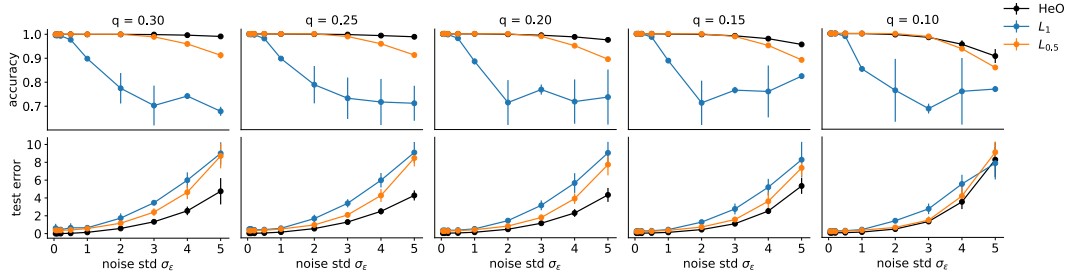

Figure 6: The variable selection of $400$-dimensional linear regressions using HeO (Alg. 6, *Appendix*), Lasso ($L_1$) regression [40] and $L_{0.5}$ regression [41]. We report the accuracy of each algorithm in determining whether each variable should be ignored for prediction and their MSE on the test set. The mean (dots) and standard deviation (bars) are estimated over 10 runs.

ceptron $\mathbf{y} = \mathbf{\Gamma}(\mathbf{v}; W_{\text{GT}}) = \text{ReLu}(W_{\text{GT}}\mathbf{v})$, where $\text{ReLu}(x) = \max\{0, x\}$, $W_{\text{GT}} \in \{-1, 0, 1\}^{m \times n}$ is the ground-truth ternary weight parameter, $\mathbf{v} \in \{-1, 0, 1\}^n$ is the input, and $\mathbf{y}$ is the model output. We aim to find the ternary configuration $W \in \{-1, 0, 1\}^{m \times n}$ minimizing the loss $\text{MSE}(W, \mathcal{D}) = \frac{1}{|\mathcal{D}|} \sum_{(\mathbf{v}, \mathbf{y}) \in \mathcal{D}} \|\mathbf{\Gamma}(\mathbf{v}; W) - \mathbf{y}\|^2$. We generalize our HeO from the binary to the ternary case by representing a ternary variable with two bits, where each element of $W$ can be represented as a function of $\mathbf{s}$ (see Alg. 5, *Appendix* for details), and the target function is defined as

$$f(\mathbf{s}) = \frac{1}{|\mathcal{D}|} \sum_{(\mathbf{v}, \mathbf{y}) \in \mathcal{D}} \|\mathbf{\Gamma}(\mathbf{v}; W(\mathbf{s})) - \mathbf{y}\|^2. \tag{16}$$

As shown in Fig. 5, HeO robustly exceeds MCGE under different dataset size $|\mathcal{D}|$ and output size $m$.

**Mixed combinatorial optimization.** In high-dimensional linear regression, usually only a small fraction of the variables significantly contribute to prediction. Identifying and selecting a subset of variables with strong predictive power—a process known as variable selection—is crucial, as it improves the generalizability and interpretability of the regression model [42]. However, direct variable selection is an NP-hard combinatorial optimization mixed with continuous variables [43]. As a practical alternative, regularization methods like Lasso algorithm are commonly employed [40].

Supposed a dataset is generated from a linear model, in which the relation between input $\mathbf{v}$ and output $y$ is $y = \boldsymbol{\beta}^* \cdot \mathbf{v} + \epsilon$, where $\boldsymbol{\beta}^*$ is the ground-truth linear coefficient and $\epsilon$ is independent Gaussian noise with standard deviation $\sigma_\epsilon$. Suppose that only a small proportion (denoted as $q \in (0, 1)$) of coordinates in $\beta_i^*$ are non-zero. Our goal is to identify these coordinates through an indicator vector $\mathbf{s} \in \{-1, 1\}^n$ (1 for selection and $-1$ for non-selection) and estimate these non-zero coefficients. The target function of the problem is ($\mathbf{1} \in \mathbb{R}^n$ is the all-one vector)

$$f(\mathbf{s}, \boldsymbol{\beta}) = \frac{1}{|\mathcal{D}|} \sum_{(\mathbf{v}, \mathbf{y}) \in \mathcal{D}} \left| \left(\boldsymbol{\beta} \odot \frac{\mathbf{s} + \mathbf{1}}{2}\right) \cdot \mathbf{v} - \mathbf{y} \right|^2. \tag{17}$$

We solve the problem through HeO (Alg. 6, *Appendix*), where we minimize the loss relative to $\boldsymbol{\theta}$ while slowing varying $\boldsymbol{\beta}$ via its error gradient. After obtaining the indicator $\mathbf{s}$, we conduct an ordinary least squares regression on the variables selected by the $\mathbf{s}$ to estimate their coefficients, and treat other variables' coefficients as zero. As shown in Fig.6, our HeO outperforms both Lasso regression and the more advanced $L_{0.5}$ regression [41] in terms of producing more accurate indicators $\mathbf{s}$ and achieving lower test prediction errors across various $q$ and $\sigma_\epsilon$ settings. Importantly, to give the variable selection prediction, our HeO does not need to know the level of $q$ and $\sigma_\epsilon$ in advance.

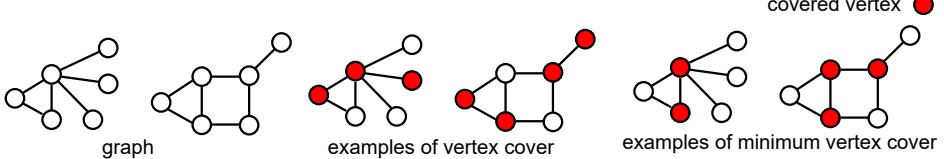

Figure 7: The illustration of minimum vertex cover.

Table 1: Attributes of graphs and the vertex cover sizes of HeO (Alg. 7, *Appendix*) and FastVC [44].

| Graph name | Vertex number | Edge number | FastVC | HeO |
| --- | --- | --- | --- | --- |
| tech-RL-caida | 190914 | 607610 | 78306 | **77372 (17)** |
| soc-youtube | 495957 | 1936748 | 149458 | **148875 (25)** |
| inf-roadNet-PA | 1087562 | 1541514 | 588306 | **587401 (104)** |
| inf-roadNet-CA | 1957027 | 2760388 | 1063352 | **1061339 (32)** |
| socfb-B-anon | 2937612 | 20959854 | 338724 | **312531 (194)** |
| socfb-A-anon | 3097165 | 23667394 | 421123 | **387730 (355)** |
| socfb-uci-uni | 58790782 | 92208195 | 869457 | **867863 (36)** |

**Constrained binary optimization.** Minimum vertex cover (MVC) is a class of the constrained combinatorial optimization which has wide applications [45], as illustrated in Fig. 7. Given an undirected graph $\mathcal{G}$ with vertex set $\mathcal{V}$ and edge set $\mathcal{E}$, the MVC is to find the minimum subset $\mathcal{V}_c \subset \mathcal{V}$, so that for each edge $e \in \mathcal{E}$, at least one of its endpoints belongs to $\mathcal{V}_c$. The target function and the constrains are expressed as

$$f(\mathbf{s}) = \sum_{i=1}^{n} \frac{s_i + 1}{2}, \quad \text{subject to } g_{ij}(\mathbf{s}) = (1 - \frac{s_i + 1}{2})(1 - \frac{s_j + 1}{2}) = 0, \quad \forall i, j, e_{ij} \in \mathcal{E}. \quad (18)$$

We combine the HeO with penalty function (Alg. 7, *Appendix*) for solving MVC. We compare our HeO with FastVC [44], a powerful MVC heuristic algorithm, on massive real world graph datasets [3]. For a fair comparison, we keep the run time of two algorithms as the same for each dataset. As shown in Tab. 1, our HeO can find smaller cover sets than that of FastVC.

## 5   Discussion

Existing model-based combinatorial optimization approaches encode the solution space via a parameterized distribution with iterative parameter updates [46]. In contrast to HeO, which requires only one sample per iteration, they necessitate a large number of samples per iteration. The Gibbs-With-Gradient algorithm [47] uses gradient information for combinatorial optimization but searches in the discrete solution space instead of the continuous one, as did HeO. Denoising diffusion model (DDM) [24] has been applied for solving combinatorial optimization problems [48]. Although the diffusion process in DDMs akin to the heat diffusion in our HeO, DDMs require a substantial data for training and necessitate reversing the diffusion process to generate data that from the target distribution. In contrast, HeO needs no training, and it is unnecessary to strictly adhering to the monotonic $\tau_t$ in the optimization process, as under different $\tau$, the function $u(\tau, \boldsymbol{\theta})$ shares the same optima with that of the original problem $h(\boldsymbol{\theta})$. This claim is empirically corroborated in Fig. S3, *Appendix*, where HeO applying non-monotonic schedules of $\tau_t$ still demonstrates superior performance.

---

[3] http://networkrepository.com/

Our HeO can be viewed as a stochastic Gaussian continuation (GC) method [49] with projection. GC has been applied for non-convex optimization, though it has not yet been used for combinatorial optimization. The optimal convexification of GC [50] underpins potentially theoretical advantages of HeO. One key distinction is that GC typically optimizes each sub-problem (corresponding to $u(\boldsymbol{\theta}, \tau_t)$) at each $t$ up to some criteria, whereas HeO merely performs a single-step gradient descent. Also, Eq. (8) corresponds to a variation of the evolution strategy, a robust optimizer for non-differentiable function [51], while HeO use a different gradient estimation (Eq. (11)), see *Appendix* for details. Additionally, our HeO is related to randomized smoothing, which has been applied to non-smooth optimization [52] or neural network regularization [53]. The distinctive feature of our HeO is that, across different $\tau$, the smoothed function $u(\tau, \boldsymbol{\theta})$ retains the optima of the original function $h(\boldsymbol{\theta})$ (Thm. 1). This distinguishes our HeO from methods based on quantum adiabatic theory [9], bifurcation theory [12] and other relaxation strategies [54], in which the optima of the smoothed function can be different from the original one [27, 55]. This is verified in Fig. S3, *Appendix*.

The heat equation in our HeO can be naturally extended to general parabolic differential equations, given that a broad spectrum of them obey the backward uniqueness [56]. For example, we can use $\partial_\tau u(\tau, \boldsymbol{\theta}) = \triangledown_{\boldsymbol{\theta}}[A \triangledown_{\boldsymbol{\theta}} u(\tau, \boldsymbol{\theta})]$ to replace the Eq. (6), where $A$ is a real positive definite matrix. Prior researches have demonstrated that the optimization procedure can be significantly accelerated by preconditioning [57] or Fisher information matrix [58], implying that choosing a proper matrix $A$ could substantially improve the efficacy of the HeO. Additionally, since $\tau_t$ does not necessarily have to be monotonic due to the cooperative optimization property of HeO, it is feasible to explore various $\tau_t$ schedules (possibly non-monotonic) to further enhance performance. Moreover, our HeO can be integrated with other existing techniques for combinatorial optimization problems with constraints, such as Augmented Lagrangian methods, to achieve better performance [59].

Despite the effectiveness of HeO on the combinatorial optimization problems from different domains we have considered, it has limitations. First, current HeO is inefficient for integer linear programming and routing problems, primarily due to that it is cumbersome to encode integer variables through the Bernoulli distribution in our framework. Nevertheless, integrating HeO with other techniques such as advanced Metropolis-Hastings algorithm [60] may path to broaden the applicability of our methodology to a wider range of combinatorial optimization problems. Besides, our HeO allows for further customization by incorporating additional terms that integrate problem-specific prior knowledge or by hybridizing with other metaheuristic algorithms, allowing for more effective exploration of the configuration space. Second, our HeO can not be theoretically guaranteed for converging to the global minimum. In general, finding the global minimum is not theoretically guaranteed for non-convex optimization problems [61], such as the combinatorial optimization problems studied in this paper. However, it can be demonstrated that the gradient of the target function under heat diffusion satisfies the inequality [22]:

$$|\triangledown_{\boldsymbol{\theta}} u(\tau, \boldsymbol{\theta})| \leq \frac{C}{\sqrt{\tau}},$$

where the constant $C$ depends on the dimension. This implies that the target function becomes weakly convex, enabling the finding of global minima and faster convergence under certain conditions [62].

## 6  Conclusion

In conclusion, grounded in the heat diffusion, we present a framework called heat diffusion optimization (HeO) to solve various combinatorial optimization problems. The heat diffusion facilitates the propagation of information from distant regions to the solver, expanding its receptive field, which in turn enhances its ability to search for global optima. Demonstrating exceptional performance across various scenarios, our HeO highlights the potential of utilizing heat diffusion to address challenges associated with navigating the solution space of combinatorial optimization.

## Acknowledgements

This work was supported by the National Science and Technology Major Project of China (No. 2018AAA0100303) and the Science & Technology Commission of Shanghai Municipality (No. 23JC1400800).

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

# Appendix

## A Proof of theorems

### A.1 Proof of Thm. 1

To prove the Thm. 1, we recall the backward uniqueness of the heat equation [23], which asserts that the initial state of a heat equation can be uniquely determined by its state at a time point $\tau$ under mild conditions, as shown in the following theorem.

**Theorem S4.** *Given two bounded function $h_1(\mathbf{x}), h_2(\mathbf{x})$ with domain on $\mathbb{R}^n$. Denote $u_1(\tau, \mathbf{x})$ and $u_2(\tau, \mathbf{x})$ as the solutions to the heat equation (Eq. (6)) with initial condition $u_1(0, \mathbf{x}) = h_1(\mathbf{x})$ and $u_2(0, \mathbf{x}) = h_2(\mathbf{x})$ respectively. If there exists a $\tau > 0$, such that*

$$u_1(\tau, \mathbf{x}) = u_2(\tau, \mathbf{x}), \quad \forall \mathbf{x} \in \mathbb{R}^n, \tag{S1}$$

*we have $h_1 = h_2$.*

Proof of the Thm. 1.

*Proof.* We first show that the global minima of $u(\tau, \boldsymbol{\theta})$ is also the global minima of $h(\boldsymbol{\theta})$. The cornerstone of the proof is Thm. S4. To utilize the backward uniqueness, we consider a reparameterization of $p(\mathbf{s}|\boldsymbol{\theta})$ by introducing a random variable $\mathbf{x} \sim \text{Unif}(0,1)^n$. It is easy to see that $\text{sgn}(\theta_i - x_i)$ obeys Bernoulli distribution with probability $\theta_i$ to be 1 and $1 - \theta_i$ to be $-1$. Replacing $s_i$ by $x_i$ in Eq. (2), we have a new expression of $h(\boldsymbol{\theta})$

$$h(\boldsymbol{\theta}) = \mathbb{E}_{p(\mathbf{x})}[f(\text{sgn}(\boldsymbol{\theta} - \mathbf{x}))], \quad \mathbf{x} \sim \text{Unif}(0,1)^n. \tag{S2}$$

Since the solution to the heat equation can be represented by the heat kernel, we have [22]

$$u(\tau, \boldsymbol{\theta}) = \mathbb{E}_{p(\mathbf{z})}[h(\boldsymbol{\theta} + \sqrt{2\tau}\mathbf{z})], \quad p(\mathbf{z}) = \mathcal{N}(\mathbf{0}, I). \tag{S3}$$

Therefore, we have

$$u(\tau, \boldsymbol{\theta}) = \mathbb{E}_{p(\mathbf{z})}[\mathbb{E}_{p(\mathbf{x})}[f(\text{sgn}(\boldsymbol{\theta} + \sqrt{2\tau}\mathbf{z} - \mathbf{x}))]] = \mathbb{E}_{p(\mathbf{x})}[\mathbb{E}_{p(\mathbf{z})}[f(\text{sgn}(\boldsymbol{\theta} - (\mathbf{x} + \sqrt{2\tau}\mathbf{z})))]]. \tag{S4}$$

Denote $u(\tau, \mathbf{x}; \boldsymbol{\theta}) = \mathbb{E}_{p(\mathbf{z})}[f(\text{sgn}(\boldsymbol{\theta} - (\mathbf{x} + \sqrt{2\tau}\mathbf{z})))], \mathbf{x} \in (0,1)^n$. Noticed that $u(\tau, \mathbf{x}; \boldsymbol{\theta})$ is the solution of the following unbounded heat equation respect to the time $\tau$ and location $\mathbf{x}$ restricted on the region $\mathbf{x} \in (0,1)^n$

$$\begin{cases} \partial_\tau u(\tau, \mathbf{x}; \boldsymbol{\theta}) &= \Delta_{\mathbf{x}} u(\tau, \mathbf{x}; \boldsymbol{\theta}), & \tau > 0, & \mathbf{x} \in \mathbb{R}^n \\ u(\tau, \mathbf{x}; \boldsymbol{\theta}) &= f(\text{sgn}(\boldsymbol{\theta} - \mathbf{x})), & \tau = 0, & \mathbf{x} \in \mathbb{R}^n \end{cases}, \tag{S5}$$

hence the latter can be considered as an extension of the former. Since $u(\tau, \mathbf{x}; \boldsymbol{\theta})$ is analytic respect to $\mathbf{x} \in (0,1)^n$ for $\tau > 0$, this extension is unique. Therefore, the value of $u(\tau, \mathbf{x}; \boldsymbol{\theta})$ on $(0,1)^n, \tau > 0$ uniquely determines the solution of Eq. (S5). Denote $\check{\boldsymbol{\theta}}$ as

$$\check{\theta}_i = \begin{cases} +\infty, & s_i^* = +1 \\ -\infty, & s_i^* = -1. \end{cases} \tag{S6}$$

Then $u(\tau, \mathbf{x}; \check{\boldsymbol{\theta}}) = f^*$, for any $\tau \geq 0$ and $\mathbf{x} \in \mathbb{R}^n$, and we have

$$u(\tau, \check{\boldsymbol{\theta}}) = \mathbb{E}_{p(\mathbf{x})}[u(\tau, \mathbf{x}; \check{\boldsymbol{\theta}})] = f^*. \tag{S7}$$

Noticed that for $\boldsymbol{\theta} \in \bar{\mathbb{R}}^n$, we have

$$u(\tau, \boldsymbol{\theta}) = \mathbb{E}_{p(\mathbf{x})}[u(\tau, \mathbf{x}; \boldsymbol{\theta})] = \mathbb{E}_{p(\mathbf{x})}[\mathbb{E}_{p(\mathbf{z})}[f(\text{sgn}(\boldsymbol{\theta} - (\mathbf{x} + \sqrt{2\tau}\mathbf{z})))]] \leq \mathbb{E}_{p(\mathbf{x})}[\mathbb{E}_{p(\mathbf{z})}[f^*]] = f^*, \tag{S8}$$

and the equality is true if and only if $u(\tau, \mathbf{x}; \boldsymbol{\theta}) = f^*$ is true for $\mathbf{x} \in \mathbb{R}^n$. Therefore, if $\hat{\boldsymbol{\theta}}$ is the one of the minimas of $u(\tau, \boldsymbol{\theta})$, and we have $u(\tau, \hat{\boldsymbol{\theta}}) \geq f^*$. Similarly, since

$$u(\tau, \hat{\boldsymbol{\theta}}) = \mathbb{E}_{p(\mathbf{x})}[u(\tau, \mathbf{x}; \hat{\boldsymbol{\theta}})] = \mathbb{E}_{p(\mathbf{x})}[\mathbb{E}_{p(\mathbf{z})}[f(\text{sgn}(\hat{\boldsymbol{\theta}} - (\mathbf{x} + \sqrt{2\tau}\mathbf{z})))]] \leq \mathbb{E}_{p(\mathbf{x})}[\mathbb{E}_{p(\mathbf{z})}[f^*]] = f^*, \tag{S9}$$

we also have $u(\tau, \hat{\boldsymbol{\theta}}) \le f^*$, hence

$$u(\tau, \mathbf{x}; \hat{\boldsymbol{\theta}}) = f^* = u(\tau, \mathbf{x}; \boldsymbol{\theta}^*), \quad \mathbf{x} \in \mathbb{R}^n. \tag{S10}$$

Due to the backward uniqueness of the heat equation, we have

$$u(0, \mathbf{x}; \hat{\boldsymbol{\theta}}) = u(0, \mathbf{x}; \boldsymbol{\theta}^*), \quad \mathbf{x} \in \mathbb{R}^n, \tag{S11}$$

that is

$$h(\hat{\boldsymbol{\theta}}) = h(\boldsymbol{\theta}^*) = f^*. \tag{S12}$$

As a result, $\hat{\boldsymbol{\theta}}$ is the one of minimas of $h(\boldsymbol{\theta})$. Conversely, using Eq. (S7), it is obviously to see that if $\hat{\boldsymbol{\theta}}$ is one of minimas of $h(\boldsymbol{\theta})$, it is also one of minimas of $u(\tau, \boldsymbol{\theta})$. $\qquad\square$

## A.2   Proof of Thm. 2

Recall Eq. (S3) and use the definition of $h(\boldsymbol{\theta})$ (see Sec. 2 of the main paper), we have

$$u(\tau, \boldsymbol{\theta}) = \mathbb{E}_{p(\mathbf{z})}[\mathbb{E}_{p(\mathbf{s}|\boldsymbol{\theta}+\sqrt{2\tau}\mathbf{z})}[f(\mathbf{s})]]. \tag{S13}$$

To construct a low-variance estimation, instead of directly using the Monte Carlo gradient estimation by sampling from $p(\mathbf{z})$ and $p(\mathbf{s}|\boldsymbol{\theta}+\sqrt{2\tau}\mathbf{z})$ for gradient estimation, we manage to integrate out the stochasticity respect to $\mathbf{z}$. Use the reparameterization Eq. (S2) and Eq. (S3), we have

$$u(\tau, \boldsymbol{\theta}) = \mathbb{E}_{p(\mathbf{x})}[\mathbb{E}_{p(\mathbf{z})}[f(\mathrm{sgn}(\boldsymbol{\theta}+\sqrt{2\tau}\mathbf{z}-\mathbf{x}))]]. \tag{S14}$$

Now we can calculate the inner term $\mathbb{E}_{p(\mathbf{z})}[f(\mathrm{sgn}(\boldsymbol{\theta}+\sqrt{2\tau}\mathbf{z}-\mathbf{x}))]$. Due to the assumption, the target function $f(\mathbf{s})$ can be written as a $K$-order multilinear polynomial of $\mathbf{s}$

$$
\begin{aligned}
f(\mathbf{s}) = a_0 &+ \sum_{i_1} a_{1,i_1} s_{i_1} + \sum_{i_1 < i_2} a_{2,i_1 i_2} s_{i_1} s_{1_2} + \sum_{i_1 < i_2 < i_3} a_{3,i_1 i_2 i_3} s_{i_1} s_{1_2} s_{i_3} + \cdots \\
&+ \sum_{i_1 < \cdots < i_K} a_{K,i_1 \ldots i_K} s_{i_1} \cdots s_{i_K}.
\end{aligned}
\tag{S15}
$$

Integrating respect to each dimension of the Gaussian integral, we have [50]

$$
\begin{aligned}
&\mathbb{E}_{p(\mathbf{z})}[f(\mathrm{sgn}(\boldsymbol{\theta}+\sqrt{2\tau}\mathbf{z}-\mathbf{x}))] \\
&= a_0 + \sum_{i_1} a_{1,i_1} \tilde{s}_{i_1} + \sum_{i_1 < i_2} a_{2,i_1 i_2} \tilde{s}_{i_1} \tilde{s}_{1_2} + \sum_{i_1 < i_2 < i_3} a_{3,i_1 i_2 i_3} \tilde{s}_{i_1} \tilde{s}_{1_2} \tilde{s}_{i_3} \\
&+ \cdots + \sum_{i_1 < \cdots < i_K} a_{K,i_1 \ldots i_K} \tilde{s}_{i_1} \cdots \tilde{s}_{i_K},
\end{aligned}
\tag{S16}
$$

where

$$\tilde{s}_i = \mathbb{E}_{p(z_i)}[\mathrm{sgn}(\theta_i + \sqrt{2\tau}z_i - x_i)] = \mathrm{erf}(\frac{\theta_i - x_i}{\sqrt{\tau}}), \tag{S17}$$

where $\mathrm{erf}(\cdot)$ is the error function. Therefore, we have

$$u(\tau, \boldsymbol{\theta}) = \mathbb{E}_{p(\mathbf{x})}[f(\mathrm{erf}(\frac{\boldsymbol{\theta} - \mathbf{x}}{\sqrt{\tau}}))], \tag{S18}$$

where $\mathrm{erf}(\cdot)$ is the element-wise error function.

## A.3   Proof of Thm. 3

*Proof.* Define the square loss of $\boldsymbol{\theta}$ as

$$e(\boldsymbol{\theta}) = (h(\boldsymbol{\theta}) - h(\boldsymbol{\theta}^*))^2. \tag{S19}$$

According to the definition of $h(\boldsymbol{\theta})$, we have

$$e(\boldsymbol{\theta}) = \mathbb{E}_{p(\mathbf{x})}[f(\mathrm{sgn}(\boldsymbol{\theta} - \mathbf{x})) - f(\mathrm{sgn}(\boldsymbol{\theta}^* - \mathbf{x}))]^2 \le \mathbb{E}_{p(\mathbf{x})}[(f(\mathrm{sgn}(\boldsymbol{\theta} - \mathbf{x})) - f(\mathrm{sgn}(\boldsymbol{\theta}^* - \mathbf{x})))^2]. \tag{S20}$$

Define the error function

$$r(\tau, \mathbf{x}; \boldsymbol{\theta}) = u(\tau, \mathbf{x}; \boldsymbol{\theta}) - u(\tau, \mathbf{x}; \boldsymbol{\theta}^*). \tag{S21}$$

Then the error function satisfies the following heat equation

$$\begin{cases} \partial_\tau r(\tau, \mathbf{x}; \boldsymbol{\theta}) &= \quad\quad \nabla_{\mathbf{x}} r(\tau, \mathbf{x}; \boldsymbol{\theta}) \\ r(0, \mathbf{x}; \boldsymbol{\theta}) &= \quad f(\mathrm{sgn}(\boldsymbol{\theta} - \mathbf{x})) - f(\mathrm{sgn}(\boldsymbol{\theta}^* - \mathbf{x})) \end{cases}. \tag{S22}$$

Define the energy function of the error function $r(\tau, \mathbf{x}; \boldsymbol{\theta})$ as

$$E(\tau; \boldsymbol{\theta}) = \int_{\mathbb{R}^n} r^2(\tau, \mathbf{x}; \boldsymbol{\theta}) p(\mathbf{x}) d\mathbf{x}. \tag{S23}$$

Then applying the heat equation and the integration by parts, we have

$$\frac{d}{d\tau} E(\tau; \boldsymbol{\theta}) = -2 \int_{\mathbb{R}^n} \|\nabla r(\tau, \mathbf{x}; \boldsymbol{\theta})\|^2 p(\mathbf{x}) d\mathbf{x}. \tag{S24}$$

Hence we have for $0 < \tau_1 < \tau_2$

$$E(\tau_1; \boldsymbol{\theta}) = E(\tau_2; \boldsymbol{\theta}) + 2 \int_{\tau_1}^{\tau_2} \int_{\mathbb{R}^n} \|\nabla r(\tau, \mathbf{x}; \boldsymbol{\theta})\|^2 p(\mathbf{x}) d\mathbf{x} d\tau. \tag{S25}$$

Use the Harnack's inequality [22], we have

$$\|\nabla r(\tau, \mathbf{x}; \boldsymbol{\theta})\|^2 \le r(\tau, \mathbf{x}; \boldsymbol{\theta}) \partial_\tau r(\tau, \mathbf{x}; \boldsymbol{\theta}) + \frac{n}{2\tau} r^2(\tau, \mathbf{x}; \boldsymbol{\theta}), \tag{S26}$$

combine with Eq. (S25), we have

$$E(\tau_1; \boldsymbol{\theta}) \le E(\tau_2; \boldsymbol{\theta}) + \frac{n}{2} \int_{\tau_1}^{\tau_2} \frac{E(\tau; \boldsymbol{\theta})}{\tau} d\tau. \tag{S27}$$

Using the Minkowski inequality on the measure $p(\mathbf{x})$, we have

$$\begin{aligned} h(\boldsymbol{\theta}) - h(\boldsymbol{\theta}^*) = e^{1/2}(\boldsymbol{\theta}) \le & \Big( \int_{\mathbb{R}^n} (f(\mathrm{sgn}(\boldsymbol{\theta} - \mathbf{x})) - u(\tau_1; \mathbf{x}; \boldsymbol{\theta}))^2 p(\mathbf{x}) d\mathbf{x} \Big)^{1/2} \\ & + \Big( \int_{\mathbb{R}^n} (u(\tau_1; \mathbf{x}; \boldsymbol{\theta}) - u(\tau_1; \mathbf{x}; \boldsymbol{\theta}^*))^2 p(\mathbf{x}) d\mathbf{x} \Big)^{1/2} \\ & + \Big( \int_{\mathbb{R}^n} (f(\mathrm{sgn}(\boldsymbol{\theta}^* - \mathbf{x})) - u(\tau_1; \mathbf{x}; \boldsymbol{\theta}^*))^2 d\mathbf{x} \Big)^{1/2} \\ = & \Big( \int_{\mathbb{R}^n} (f(\mathrm{sgn}(\boldsymbol{\theta} - \mathbf{x})) - u(\tau_1; \mathbf{x}; \boldsymbol{\theta}))^2 p(\mathbf{x}) d\mathbf{x} \Big)^{1/2} \\ & + \Big( \int_{\mathbb{R}^n} (f(\mathrm{sgn}(\boldsymbol{\theta}^* - \mathbf{x})) - u(\tau_1; \mathbf{x}; \boldsymbol{\theta}^*))^2 d\mathbf{x} \Big)^{1/2} + E^{1/2}(\tau_1; \boldsymbol{\theta}). \end{aligned} \tag{S28}$$

Recall the continuity of the heat equation:

$$\lim_{\tau \to 0} \int_{\mathbb{R}^n} (u(\tau, \mathbf{x}; \boldsymbol{\theta}) - f(\mathrm{sgn}(\boldsymbol{\theta} - \mathbf{x})))^2 p(\mathbf{x}) d\mathbf{x} = 0. \tag{S29}$$

Therefore, given $\epsilon > 0$, there exists a $\tau_1 > 0$, such that

$$\begin{aligned} & \Big( \int_{\mathbb{R}^n} (f(\mathrm{sgn}(\boldsymbol{\theta} - \mathbf{x})) - u(\tau_1; \mathbf{x}; \boldsymbol{\theta}))^2 p(\mathbf{x}) d\mathbf{x} \Big)^{1/2} \\ & + \Big( \int_{\mathbb{R}^n} (f(\mathrm{sgn}(\boldsymbol{\theta}^* - \mathbf{x})) - u(\tau_1; \mathbf{x}; \boldsymbol{\theta}^*))^2 p(\mathbf{x}) d\mathbf{x} \Big)^{1/2} < \epsilon. \end{aligned} \tag{S30}$$

Recall Eq. (S27), we then have the error control for $e(\boldsymbol{\theta})$:

$$e^{1/2}(\boldsymbol{\theta}) \le E^{1/2}(\tau_1; \boldsymbol{\theta}) + \epsilon \le \Big( E(\tau_2; \boldsymbol{\theta}) + \frac{n}{2} \int_{\tau_1}^{\tau_2} \frac{E(\tau; \boldsymbol{\theta})}{\tau} d\tau \Big)^{1/2} + \epsilon. \tag{S31}$$

Noticed that

$$E(\tau; \boldsymbol{\theta}) \le (\breve{f} - f^*) \mathbb{E}_{p(\mathbf{x})} [u(\tau, \mathbf{x}; \boldsymbol{\theta}) - u(\tau, \mathbf{x}; \boldsymbol{\theta}^*)] = (\breve{f} - f^*)(u(\tau, \boldsymbol{\theta}) - u(\tau, \boldsymbol{\theta}^*)), \tag{S32}$$

where $\breve{f} = \max_{\mathbf{s}} f(\mathbf{s})$, and we prove the theorem. $\quad\square$

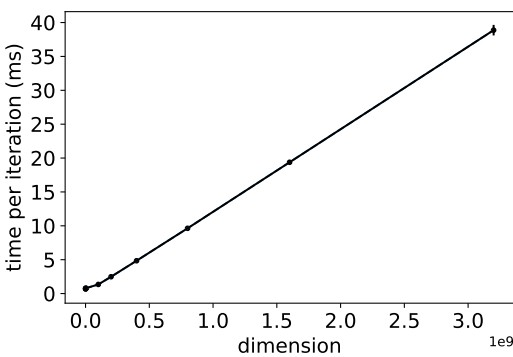

Figure S1: The time cost per iteration (ms) of the HeO framework increases linearly with the dimensionality of the problem. We present the results averaged over five tests, with error bars representing three standard deviations.

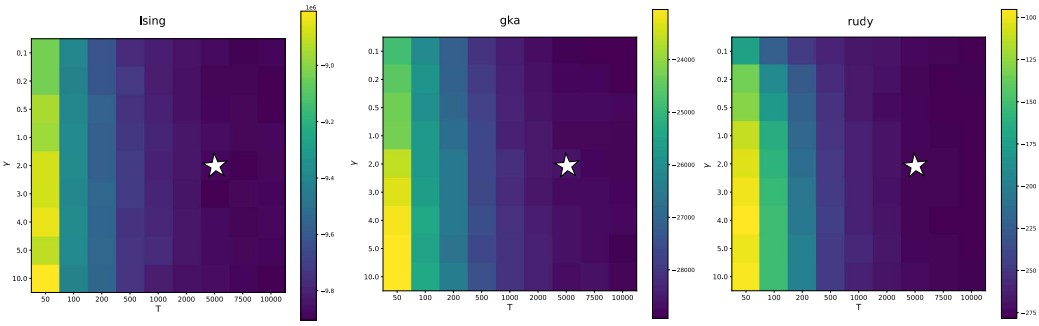

Figure S2: Mean result target function values (lower is better) of our HeO on three datasets (Ising, gka, and rudy) under various step sizes ($\gamma$) and iteration counts ($T$). The star denotes the settings used in Figure 3 of the main paper.

## B    Complexity analysis

We empirically estimate the time cost per iteration of our HeO for problems of different dimension $n$ in Fig. S1.

## C    Parameter sensitivity analysis

We report the performance of our HeO on a wide range of parameter settings in Fig. S2.

## D  Relation to evolution strategy

We show that the update in Eq. (8) is equivalent to a variation of the evolution strategy (ES) which is a robust and powerful algorithm for solving black-box optimization problem [51]. In fact, we have

$$
\begin{aligned}
\triangledown_{\boldsymbol{\theta}} u(\tau, \boldsymbol{\theta}) &= \triangledown_{\boldsymbol{\theta}} \mathbb{E}_{p(\mathbf{z})}[h(\boldsymbol{\theta} + \sqrt{2\tau}\mathbf{z})] \\
&= \triangledown_{\boldsymbol{\theta}} \int \frac{1}{\sqrt{4\pi\tau}^n} \exp(-\frac{1}{4\tau} \|\mathbf{z} - \boldsymbol{\theta}\|^2) h(\mathbf{z}) d\mathbf{z} \\
&= \frac{1}{2\tau} \int \frac{1}{\sqrt{4\pi\tau}^n} \exp(-\frac{1}{4\tau} \|\mathbf{z} - \boldsymbol{\theta}\|^2) h(\mathbf{z})(\mathbf{z} - \boldsymbol{\theta}) d\mathbf{z} \\
&= \frac{1}{2\tau} \int \frac{1}{\sqrt{2\pi}^n} \exp(-\frac{1}{2} \|\mathbf{z}\|^2) h(\boldsymbol{\theta} + \sqrt{2\tau}\mathbf{z}) \mathbf{z} d\mathbf{z} \\
&= \frac{1}{2\tau} \mathbb{E}_{p(\mathbf{z})}[h(\boldsymbol{\theta} + \sqrt{2\tau}\mathbf{z})\mathbf{z}].
\end{aligned}
\tag{S33}
$$

The random vector $\mathbf{z}$ corresponds to the stochastic mutation and $h(\boldsymbol{\theta} + \sqrt{2\tau}\mathbf{z})$ corresponds to the fitness in Alg. 1 in [63]. In ES, the standard deviation $2\tau$ is fixed in general, while in our HeO $\sqrt{2\tau_t}$ is varying across time. As shown in Thm. 3, the varying $\tau_t$ in our HeO offers a theoretical benefit that it controls the upper bound of the optimization result. In contrast, a constant $\tau$ in ES does not provide this benefit. Another difference between our HeO and ES is that we integral out $\mathbf{z}$ and using Eq. (11) to estimate the gradient, while in ES the gradient is estimated based on sampling $\mathbf{z}$.

## E  Relation to denoising diffusion models

Our approach, while bearing similarities to the DDM—a highly regarded and extensively utilized artificial generative model [24] that relies on the reverse diffusion process for data generation—differs in key aspects. The DDM necessitates reversing the diffusion process to generate data that from the target distribution. In contrast, it is unnecessary for our HeO to strictly adhering to the reverse time sequence $\tau_t$ in the optimization process, as under different $\tau$, the function $u(\tau, \boldsymbol{\theta})$ shares the same optima with that of the original problem $h(\boldsymbol{\theta})$. This claim is corroborated in Fig. S3. as shown below, where HeO applying non-monotonic schedules of $\tau_t$ still demonstrates superior performance. Hence, it is possible to explore diverse $\tau_t$ schedules to further performance enhancement.

## F  Cooperative optimization

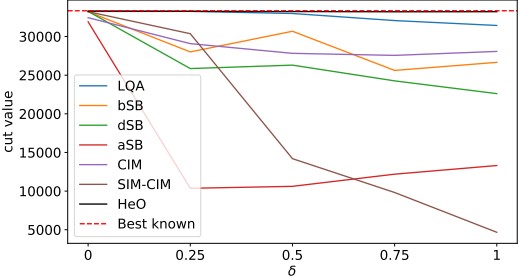

Figure S3: **Verifying the cooperative optimization mechanism of HeO.** The best cut value over 10 runs for each algorithm on the K-2000 problem [14] when the control parameters are randomly perturbed by different random perturbation level $\delta$. The red dash line is the best cut value ever find.

Our study suggests that HeO exhibits a distinct cooperative optimization mechanism, setting it apart from current methodologies. Specifically, HeO benefits from the fact that target functions $u(\tau, \boldsymbol{\theta})$ share the same optima as the original problem $h(\boldsymbol{\theta})$ for any $\tau > 0$. This characteristic allows the solver to transition between different $\tau$ values during the optimization process, eliminating the necessity for a monotonic $\tau_t$ schedule. In contrast, traditional methods such as those based on quantum adiabatic theory or bifurcation theory require a linear increase of a control parameter $a_t$ from 0 to 1. This parameter is analogous to $\tau_t$ in the HeO framework.

To empirically verify the above claim, we introduce a random perturbation to the $\tau$ schedule in Alg. 1, rendering it non-monotonic: $\tilde{\tau}_t = c_t^2 \tau_t$, where $c_t$ is uniformly distributed on $[1 - \delta, 1 + \delta]$ with $\delta$ controlling the amplitude of the perturbation. For other methods based on quantum adiabatic theory or bifurcation theory, we correspondingly introduce the perturbation as $\tilde{a}_t = \mathrm{clamp}(c_t a_t, 0, 1)$. If an algorithm contains cooperative optimization mechanism, it still works well even when the control parameter is not monotonic, as optimizing the transformed problems under different control parameters cooperatively contributes to optimizing the original problem. As shown in Fig. S3, the performance of other methods are all dramatically deteriorated. In contrast, HeO shows no substantial decline in performance, corroborating that HeO employs a cooperative optimization mechanism.

## G   Implementation details

All the experiments are conducted on a single NVIDIA RTX-3090 GPU (24GB) and an Intel(R) Xeon(R) Gold 6226R CPU (2.90GHz). For convenience, we denote $\sqrt{\tau}_t = \sigma_t$.

**Monte Carlo gradient estimation for combinatorial optimization.**   Based on Eq. (5), we construct a combinatorial optimization algorithm using the Monte Carlo gradient estimation (MCGE), as shown in Alg. 2. Noticed that we clamp the parameter $\boldsymbol{\theta}_t$ in the $[0, 1]$ for numerical stability; additionally, we binarize the $\boldsymbol{\theta}_T$ to obtain the optimized binary configuration $\mathbf{s}_T$ in the end.

---

**Algorithm 2** Monte Carlo gradient estimation for combinatorial optimization (MCGE)

---

**Input:** target function $f(\cdot)$, step size $\gamma$, sample number $M$, iteration number $T$
initialize elements of $\boldsymbol{\theta}_0$ as 0.5
**for** $t = 0$ **to** $T - 1$ **do**
    sample $\mathbf{s}^{(m)} \sim_{i.i.d.} p(\mathbf{s}|\boldsymbol{\theta}_t), \quad m = 1, \ldots, M$
    $\mathbf{g}_t \leftarrow \frac{1}{M} \sum_{m=1}^{M} f(\mathbf{s}^{(m)}) \nabla_{\boldsymbol{\theta}_t} \log p(\mathbf{s}^{(m)}|\boldsymbol{\theta}_t)$
    $\boldsymbol{\theta}_{t+1} \leftarrow \mathrm{clamp}(\boldsymbol{\theta}_t - \gamma \mathbf{g}_t, 0, 1)$
**end for**
$\mathbf{s}_T \leftarrow \mathrm{sgn}(\boldsymbol{\theta}_T - 0.5)$
**Output:** binary configuration $\mathbf{s}_T$

---

**Gradient descent with momentum.**   We provide HeO with momentum in Alg. 3.

---

**Algorithm 3** Heat diffusion optimization (HeO) with momentum

---

**Input:** target function $f(\cdot)$, step size $\gamma$, momentum $\kappa$, $\sigma$ schedule $\{\sigma_t\}$, iteration number $T$, set $\mathbf{g}_{-1} = \mathbf{0}$
initialize elements of $\boldsymbol{\theta}_0$ as 0.5
**for** $t = 0$ **to** $T - 1$ **do**
    sample $\mathbf{x}_t \sim \mathrm{Unif}[0, 1]^n$
    $\mathbf{w}_t \leftarrow \nabla_{\boldsymbol{\theta}_t} f(\mathrm{erf}(\frac{\boldsymbol{\theta}_t - \mathbf{x}_t}{\sigma_t}))$
    $\mathbf{g}_t \leftarrow \kappa \mathbf{g}_{t-1} + \gamma \mathbf{w}_t$
    $\boldsymbol{\theta}_{t+1} \leftarrow \mathrm{Proj}_{\mathcal{I}}(\boldsymbol{\theta}_t + \mathbf{g}_t)$
**end for**
$\mathbf{s}_T \leftarrow \mathrm{sgn}(\boldsymbol{\theta}_T - 0.5)$
**Output:** binary configuration $\mathbf{s}_T$

---

**Toy example.**   We set the momentum $\kappa = 0.9999$, learning rate $\gamma = 2$ and iteration number $T = 5000$ for HeO (Alg. 3). For MCGE without momentum, we set $T = 50000$, we set $\gamma = $1e-6, momentum $\kappa = 0$, and $M = 10$. For MCGE with momentum, we set momentum as 0.9.

**Max-cut problem.**   For solving the max-cut problems from the Biq Mac Library [36], we set the steps $T = 5000$ for all the algorithms. For HeO, we set $\gamma = 2$ and $\sigma_t$ linearly decreases from 1 to 0 for HeO, and we set momentum as zero. For LQA and SIM-CIM, we use the setting in [10]. For bSB, dSB, aSB, and CIM, we apply the settings in [64]. To reduce the fluctuations of the results, for each algorithm alg and instant $i$, the relative loss is calculated as $|C^{i,\mathrm{alg}} - C^i_{\min}|/|C^i_{\min}|$, where $C^{i,\mathrm{alg}}$ is the lowest output of the algorithm alg on the instance $i$ over 10 tries, and $C^i_{\min}$ is the lowest output of all 7 the algorithm on the instance $i$. For each test, we estimate the mean and std from 10 runs.

**3-SAT problem.** For Boolean 3-satisfiability (3-SAT) problem, we set the momentum $\kappa = 0.9999$, $T = 5000$, $\gamma = 2$, and $\sigma_t$ linearly decreases from $\sqrt{2}$ to 0 for HeO. According to the empirical finding that high-order loss function leads to better results [16], we include higher order terms in the target function. However, since

$$(\frac{1 - c_i s_i}{2})^k = \frac{1 - c_i s_i}{2} \tag{S34}$$

for any $s_i, c_i \in \{-1, 1\}$, directly introducing higher order term in the $f(\mathbf{s})$ is useless. Instead, we adjust the gradient from

$$\bigtriangledown_{\boldsymbol{\theta}_t} f(\text{erf}(\frac{\boldsymbol{\theta}_t - \mathbf{x}_t}{\sigma_t}))) = \bigtriangledown_{\boldsymbol{\theta}_t} [\sum_{h=1}^{H} \prod_{i=1}^{3} \frac{1}{2}(1 - c_{h_i}(\text{erf}(\frac{\boldsymbol{\theta}_t - \mathbf{x}_t}{\sigma_t}))_{h_i})], \tag{S35}$$

to

$$\bigtriangledown_{\boldsymbol{\theta}_t} [\sum_{h=1}^{H} \prod_{i=1}^{3} (\frac{1}{2}(1 - c_{h_i}(\text{erf}(\frac{\boldsymbol{\theta}_t - \mathbf{x}_t}{\sigma_t}))_{h_i}))^4]. \tag{S36}$$

We consider the 3-SAT problems with various number of variables from the SATLIB [37]. For each number of variables in the dataset, we consider the first 100 instance. We apply the same configuration of that in [16] for both 3-order solver and 2-order oscillation Ising machine solver. The energy gap of the 2-order solver is set as 1. For each test, we estimate the mean and std from 100 runs.

---

**Algorithm 4** Heat diffusion optimization (HeO) for 3-SAT problem

**Input:** adjusted target function $f(\mathbf{s}) = \sum_{h=1}^{H} \prod_{i=1}^{3} \left(\frac{1 - c_{h_i} s_{h_i}}{2}\right)^4$ (Eq. (S36)), step size $\gamma$, momentum $\kappa$, $\sigma$ schedule $\{\sigma_t\}$, iteration number $T$
initialize elements of $\boldsymbol{\theta}_0$ as 0.5, set $\mathbf{g}_{-1} = \mathbf{0}$
**for** $t = 0$ **to** $T - 1$ **do**
    sample $\mathbf{x}_t \sim \text{Unif}[0, 1]^n$
    $\mathbf{w}_t \leftarrow \bigtriangledown_{\boldsymbol{\theta}_t} f(\text{erf}(\frac{\boldsymbol{\theta}_t - \mathbf{x}_t}{\sigma_t}))$
    $\mathbf{g}_t \leftarrow \kappa \mathbf{g}_{t-1} + \gamma \mathbf{w}_t$
    $\boldsymbol{\theta}_{t+1} \leftarrow \text{Proj}_{\mathcal{I}}(\boldsymbol{\theta}_t - \gamma \mathbf{g}_t)$
**end for**
$\mathbf{s}_T \leftarrow \text{sgn}(\boldsymbol{\theta}_T - 0.5)$
**Output:** binary configuration $\mathbf{s}_T$

---

**Ternary-value neural network learning.** We represent a ternary variable $s_t \in \{-1, 0, 1\}$ as $s_t = \frac{1}{2}(s_{b,1} + s_{b,2})$ with two bits $s_{b,1}, s_{b,2} \in \{-1, 1\}$. In this way, each element of $W$ can be represented as a function of $\mathbf{s} \in \mathbb{R}^{m \times n \times 2}$. We denote this relation as a matrix-value function $W = W(\mathbf{s})$

$$W_{ij}(\mathbf{s}) = \frac{1}{2}(s_{ij,1} + s_{ij,2}), \quad i = 1, \ldots, m, \quad , j = 1, \ldots, n, \tag{S37}$$

Based on the above encoding procedure, we design the training algorithm for based on HeO in Alg. 5. The input $\mathbf{v}$ of the dataset $\mathcal{D}$ is generated from the uniform distribution on $\{-1, 0, 1\}^n$. For HeO, we set $T = 10000$, $\gamma = 0.5$, $\kappa = 0.999$, and $\sigma_t$ linearly decreasing from $\sqrt{2}$ to 0. For MCGE, we set $T = 10000$, $\gamma = 1e - 7$, $M = 10$, and $\kappa = 0.9999$. We empirically find that MCGE need high sampling number ($M$) and low learning rate ($\gamma$) for stability, while this is not the case for HeO. For each test, we estimate the mean and std from 10 runs.

**Variable selection problem.** We construct an algorithm for variable selection problem based on HeO as shown in Alg. 6, where the function $f(\mathbf{s}, \boldsymbol{\beta})$ is defined in Eq. (17).

We randomly generate 400-dimensional datasets with 1000 training samples. The input $\mathbf{v}$ is sampled from a standard Gaussian distribution. The element of the ground-truth coefficient $\boldsymbol{\beta}^*$ is uniformly distributed on $[-2, -1] \cup [1, 2]$, and each element has $1 - q$ probability of being set as zero and thus should be ignored for the prediction. We apply a five-fold cross-validation for all of methods. For our HeO, we set $T = 2000$ and $\gamma = 1$, $\kappa = 0.999$. We generate an ensemble of indicators $\mathbf{s}$ of size 100.

---

**Algorithm 5** Heat diffusion optimization (HeO) for training ternary-value neural network

---

**Input:** dataset $\mathcal{D}$, step size $\gamma$, momentum $\kappa$, $\sigma$ schedule $\{\sigma_t\}$, iteration number $T$
initialize elements of $\boldsymbol{\theta}_0$ as 1, initialize elements of $\tilde{\boldsymbol{\beta}}_0$ as 0, set $\mathbf{g}_{-1} = \mathbf{0}$
**for** $t = 0$ **to** $T - 1$ **do**
  sample $\mathbf{x}_t \sim \mathrm{Unif}[0,1]^n$
  $W_t \leftarrow W(\mathrm{erf}(\frac{\boldsymbol{\theta}_t - \mathbf{x}_t}{\sigma_t}))$ (Eq. (S37))
  $\mathrm{MSE} \leftarrow \frac{1}{|\mathcal{D}|} \sum_{(\mathbf{v},\mathbf{y}) \in \mathcal{D}} \|\mathbf{\Gamma}(\mathbf{v}; W_t) - \mathbf{y}\|^2$
  $\mathbf{w}_t \leftarrow \nabla_{\boldsymbol{\theta}_t} \mathrm{MSE}$
  $\mathbf{g}_t \leftarrow \kappa \mathbf{g}_{t-1} + \gamma \mathbf{w}_t$
  $\boldsymbol{\theta}_{t+1} \leftarrow \mathrm{Proj}_{\mathcal{I}}(\boldsymbol{\theta}_t - \mathbf{g}_t)$
**end for**
$\mathbf{s}_T = \mathrm{sgn}(\theta_T - 0.5)$
**Output:** $W_T = W(\mathbf{s}_T)$

---

---

**Algorithm 6** Heat diffusion optimization (HeO) for linear regression variable selection

---

**Input:** dataset $\mathcal{D}$, step size $\gamma$, momentum $\kappa$, $\sigma$ schedule $\{\sigma_t\}$, iteration number $T$
initialize elements of $\boldsymbol{\theta}_0$ as 1, initialize elements of $\tilde{\boldsymbol{\beta}}_0$ as 0, set $\mathbf{g}^{\beta}_{-1} = \mathbf{g}^{\theta}_{-1} = \mathbf{0}$.
**for** $t = 0$ **to** $T - 1$ **do**
  sample $\mathbf{x}^{\theta}_t \sim \mathrm{Unif}[0,1]^n$
  $\mathbf{w}^{\beta}_t \leftarrow \nabla_{\tilde{\boldsymbol{\beta}}_t} f(\mathrm{erf}(\frac{\boldsymbol{\theta}_t - \mathbf{x}_t}{\sigma_t}), \boldsymbol{\beta})$
  $\mathbf{g}^{\beta}_t \leftarrow \kappa \mathbf{g}^{\beta}_{t-1} + \frac{\gamma}{T} \mathbf{w}^{\beta}_t$
  $\tilde{\boldsymbol{\beta}}_{t+1} \leftarrow \tilde{\boldsymbol{\beta}}_t - \mathbf{g}^{\beta}_t$
  $\mathbf{w}^{\theta}_t \leftarrow \nabla_{\boldsymbol{\theta}_t} f(\mathrm{erf}(\frac{\boldsymbol{\theta}_t - \mathbf{x}_t}{\sigma_t}), \boldsymbol{\beta})$
  $\mathbf{g}^{\theta}_t \leftarrow \kappa \mathbf{g}^{\theta}_{t-1} + \gamma \mathbf{w}^{\theta}_t$
  $\boldsymbol{\theta}_{t+1} \leftarrow \mathrm{Proj}_{\mathcal{I}}(\boldsymbol{\theta}_t - \gamma \mathbf{g}^{\theta}_t)$
**end for**
$\mathbf{s}_T \leftarrow \mathrm{sgn}(\boldsymbol{\theta}_T - 0.5)$
**Output:** $\mathbf{s}_T$

---

For each $\mathbf{s}$ in the ensemble, we fit a linear model by implementing an OLS on the non-zero variables indicated by $\mathbf{s}$ and calculate the average MSE loss of the linear model on the cross-validation sets. We then select the linear model with lowest MSE on the validate sets as the output linear model. For Lasso and $L_{0.5}$ regression, we follow the implementation in [41] with 10 iterations. the regularization parameter is selected by cross-validation from $\{0.05, 0.1, 0.2, 0.5, 1, 2, 5\}$. For each test, we estimate the mean and std from 10 runs.

Table S2: **The attributes of the real world graphs and the parameter settings of HeO.**

| graph name | $|V|$ | $|E|$ | $T$ | $\gamma$ | $\lambda_t$ | $\sigma_t$ |
|---|---|---|---|---|---|---|
| tech-RL-caida | 190914 | 607610 | 200 | 2.5 | linearly from 0 to 2.5 | |
| soc-youtube | 495957 | 1936748 | 200 | 2.5 | linearly from 0 to 2.5 | |
| inf-roadNet-PA | 1087562 | 1541514 | 200 | 2.5 | linearly from 0 to 7.5 | |
| inf-roadNet-CA | 1957027 | 2760388 | 200 | 5 | linearly from 0 to 7.5 | linearly from $\sqrt{2}$ to 0 |
| socfb-B-anon | 2937612 | 20959854 | 50 | 2.5 | linearly from 0 to 5 | |
| socfb-A-anon | 3097165 | 23667394 | 50 | 2.5 | linearly from 0 to 5 | |
| socfb-uci-uni | 58790782 | 92208195 | 50 | 2.5 | linearly from 0 to 5 | |

**Minimum vertex cover problem.** For constrained binary optimization

$$\min_{\mathbf{s} \in \{1,1\}^n} f(\mathbf{s}), \tag{S38}$$

$$g_k(\mathbf{s}) \leq 0, \quad , k = 1, \ldots, K, \tag{S39}$$

we put the constrains as the penalty function with coefficient $\lambda$ into the target function

$$f_\lambda(\mathbf{s}) = f(\mathbf{s}) - \lambda \sum_{k=1}^{K} g_k(\mathbf{s}), \tag{S40}$$

and the corresponding algorithm is shown in Alg. 7.

---

**Algorithm 7** Heat diffusion optimization (HeO) for constrained binary optimization

---

**Input:** target function with penalty $f_\lambda$, step size $\gamma$, $\sigma$ schedule $\{\sigma_t\}$, penalty coefficients schedule $\{\lambda_t\}$, iteration number $T$
initialize elements of $\boldsymbol{\theta}_0$ as 0.5, set $\mathbf{g}_{-1} = \mathbf{0}$.
**for** $t = 0$ **to** $T - 1$ **do**
    sample $\mathbf{x}_t$ from $\mathrm{Unif}[0, 1]^n$
    $\mathbf{w}_t \leftarrow \nabla_{\boldsymbol{\theta}_t} f_{\lambda_t}(\frac{\boldsymbol{\theta}_t - \mathbf{x}_t}{\sigma_t})$
    $\mathbf{g}_t \leftarrow \kappa \mathbf{g}_{t-1} + \gamma \mathbf{w}_t$
    $\boldsymbol{\theta}_{t+1} \leftarrow \mathrm{Proj}_{\mathcal{I}}(\boldsymbol{\theta}_t - \gamma \mathbf{g}_t)$
**end for**
$\mathbf{s}_T \leftarrow \mathrm{sgn}(\boldsymbol{\theta}_T - 0.5)$
**Output:** $\mathbf{s}_T$

---

---

**Algorithm 8** Refinement of the result of MVC

---

**Input:** the result of HeO $\mathbf{s}_T$
**for** $i = 1$ **to** $n$ **do**
    set $s_{T,i}$ as 0 if $\mathbf{s}_T$ is still a vertex cover
**end for**
**Output:** $\mathbf{s}_T$

---

We implement the HeO on a single NVIDIA RTX 3090 GPU for all the minimum vertex cover (MVC) experiments. Let $\mathbf{s}$ be the configuration to be optimized, in which $s_i$ is 1 if we select $i$-th vertex into $\mathcal{V}_c$, otherwise we do not select $i$-vertex into $\mathcal{V}_c$. The target function to be minimize is the size of $\mathcal{V}_c$: $f(\mathbf{s}) = \sum_{i=1}^{n} \frac{s_i+1}{2}$, and the constrains are

$$g_{ij}(\mathbf{s}) = (1 - \frac{s_i + 1}{2})(1 - \frac{s_j + 1}{2}) = 0, \quad \forall i, j, e_{ij} \in \mathcal{E}, \tag{S41}$$

where $e_{ij}$ represent the edge connecting the $i$ and $j$-th vertices. We construct the target function $f_\lambda(\mathbf{s}) = f(\mathbf{s}) + \lambda \sum_{e_{ij} \in \mathcal{E}} g_{ij}(\mathbf{s})$. The term with the positive factor $\lambda$ penalizes vector $\mathbf{s}$ when there are uncovered edges. After the HeO outputs the result $\mathbf{s}_T$, we empirically find that its subset may also form a vertex cover for the graph $\mathcal{G}$, so we implement the following refinement on the result $\mathbf{s}_T$, as shown in Alg. 8. We report the vertex number, edge number and settings of HeO in Tab. S2. For FastVC, we follow the settings in [44] and use its codebase, and set the cut-off time as the same as the time cost of HeO. For each test, we estimate the mean and std from 10 runs.

