# OpenReview forum: "Efficient Combinatorial Optimization via Heat Diffusion"
_NeurIPS.cc/2024/Conference — NeurIPS 2024 poster_

### Official Review · Reviewer_VgKr · 2024-07-13

**Soundness:** 3
**Presentation:** 3
**Contribution:** 2
**Rating:** 6
**Confidence:** 3

**Summary:**

This work solves combinatorial optimization problems using the gradient method by transforming the discrete problem into a continuous problem. Under the invariant of the optimal solution, the authors transformed the hard continuous problem into an easier problem by changing the objective function using a heating equation and improved the calculation process of the gradient which makes the problem more tractable.

**Strengths:**

1. The authors established the basic theory for the method proposed in the paper.
2. Extensive experients have been conducted and clearly figures have been presented.

**Weaknesses:**

1. The motivation of this paper seems to improve the scope of the search-based combinatorial optimization solver, but the proposed method is to change the objective function to improve the efficiency of the gradient method. The method seems a little bit irrelevant to the original motivation of the paper.
2. The paper does not fully discuss the combinatorial optimization problem with constraints ( only do some experiments on the minimum vertex cover problem ), and the description of the violation of constraints is not sufficient.

**Questions:**

1. The discussion of equations (8) to (9) is a little confusing.  Could you please explain how to calculate the projection map in equation(9)?

**Limitations:**

1. The method does not fully discuss the combinatorial problem with constraints which is the main part of the combinatorial problem.

---

> ### Author Rebuttal · Authors · 2024-08-03
>
> > 1. The motivation of this paper seems to improve the scope of the search-based combinatorial optimization solver, but the proposed method is to change the objective function to improve the efficiency of the gradient method. The method seems a little bit irrelevant to the original motivation of the paper.
>
> Thank you for pointing this out. We will clarify this in the revised version. The primary reason we focus on gradient methods is that, to address the limitations of search-based combinatorial optimization solvers, we reformulated the combinatorial problems into continuous optimization problems. This reformulation allows us to apply heat diffusion to propagate information across the configuration space. Consequently, gradient methods are a natural choice for solving these continuous optimization problems.
>
> Additionally, gradient methods can be viewed as a specific type of search-based combinatorial optimization solver, where the gradient provides local information that guides the solver such as Gibbs sampling methods in the direction of improving the solution [1].
>
> > 2. The paper does not fully discuss the combinatorial optimization problem with constraints (only do some experiments on the minimum vertex cover problem), and the description of the violation of constraints is not sufficient.
>
> Thank you for pointing this out. We will clarify this in the revised version. In Section 4 of the main paper, we present experiments on large-scale instances of the minimum vertex cover problem, where our HeO consistently finds good solutions without violating any constraints. These instances involve millions of nonlinear constraints (see Eq. 18), with edge and vertex counts reaching up to 50 million (see Table 1). Our experiments demonstrate HeO's ability to handle complex, large-scale problems with a massive number of nonlinear constraints. Additionally, HeO can be integrated with other existing techniques for combinatorial optimization problems with constraints, such as Augmented Lagrangian methods, to achieve better performance [2].
>
>
> > 3. The discussion of equations (8) to (9) is a little confusing. Could you please explain how to calculate the projection map in equation(9)?
>
> Thank you for pointing this out. We will clarify this in the revised version. The projection of a point $\mathbf{x} \in \mathbb{R}^n$ onto the region $\mathcal{I}=[0,1]^n$ is calculated for each coordinate $i$ of $x$ as:
> \begin{align}
> \mathrm{Proj}_{\mathcal{I}}[x]_i =\min(1, \max(0, x_i)).
> \end{align}
>
> [1] Grathwohl W, Swersky K, Hashemi M, et al. Oops i took a gradient: Scalable sampling for discrete distributions[C]//International Conference on Machine Learning. PMLR, 2021: 3831-3841.
>
> [2] Birgin E G, Martínez J M. Practical augmented Lagrangian methods for constrained optimization[M]. Society for Industrial and Applied Mathematics, 2014.

---

### Official Review · Reviewer_uVQ4 · 2024-07-13

**Soundness:** 2
**Presentation:** 3
**Contribution:** 3
**Rating:** 5
**Confidence:** 3

**Summary:**

This paper proposes the Heat Diffusion Optimization (HeO) method, which leverages thermodynamic principles to enhance combinatorial optimization (CO) problems. Specifically, it integrates heat diffusion equations into gradient-based optimization to improve efficiency and help escape local minima.

**Strengths:**

1. The paper is generally well-written, with a good balance of examples, explanations, and discussions.
2. The idea of using heat diffusion to propagate information across the solution space is novel and interesting for solving CO problems.
3. The proposed method is validated on different types of CO problems with varying scales.

**Weaknesses:**

1. While the work introduces a new approach for gradient-based combinatorial optimization, it inherits the limitations of gradient-based methods. It may struggle with complex CO problems, such as routing problems, as discussed by the authors.

2. The paper could benefit from a deeper theoretical and empirical analysis of the HeO algorithm. For example, a detailed analysis of the convergence properties and computational complexity of the algorithm is needed. Also, summarizing and providing recommendations on parameter sensitivity and selection would be valuable.

3. The paper demonstrates improved performance over several classic solvers like MDGE, SA, and LQA. However, it does not provide enough evidence or discuss the potential of HeO to achieve state-of-the-art performance on the studied CO problems.

4. The submission seems to be not incomplete as there lack of appendices or supplementary materials that verify the soundness of the theorems presented in Section 3.

Overall, I feel like this paper is proposing a very interesting new method for CO with potential. It would benefit greatly from a major revision that includes more details, proofs, and broader experiments.

**Questions:**

None.

**Limitations:**

No concerns.

---

> ### Author Rebuttal · Authors · 2024-08-03
>
> > 1. The paper could benefit from a deeper theoretical and empirical analysis of the HeO algorithm. For example, a detailed analysis of the convergence properties and computational complexity of the algorithm is needed. Also, summarizing and providing recommendations on parameter sensitivity and selection would be valuable.
>
>
> Thank you for the valuable suggestions.
>
> For **convergence analysis**, we will include a theoretical discussion in the main paper. In general, finding the global minimum is not theoretically guaranteed for non-convex optimization problems [1], such as the combinatorial optimization problems studied in this paper. However, it can be demonstrated that the gradient of the target function under heat diffusion satisfies the inequality [2]:
>
> \begin{align}
> \left\| \bigtriangledown_{{\theta}}u(\tau,{\theta} )\right\| \leq \frac{C}{\sqrt{\tau}},
> \end{align}
>
> where $C$ is a constant depended on the dimension and $||f||_{\infty}$. This suggests that the target function becomes weakly convex, which has been shown to enable the discovery of global minima and achieve faster convergence rates under certain conditions [3].
>
>
> For **complexity analysis**, we will add a discussion on computational complexity in the main paper. The complexity of our algorithm is relatively low. At each step, the most computationally intensive operation is calculating the gradient of the target function $h(\theta)$. This can sometimes be expressed explicitly or be efficiently computed using automatic differentiation tools like PyTorch's autograd. The overall computational time is primarily dependent on the number of iterations $T$. As demonstrated in Figure 1 in the **PDF file of the global author rebuttal**, the time cost per iteration of our methods increases linearly with the problem dimension, with a small constant coefficient. This confirms the efficiency of our method. We will include this result in the Supplementary Information after revision.
>
> For **parameter sensitivity and selection**, we will include a parameter analysis section in the supplementary information. We analyze the effects of two main parameters: the step size $\gamma$ and the number of iterations $T$ on the performance of our HeO method. Since revisions are not allowed during the rebuttal phase, we present the results in Figure 2 in the ***PDF file of the global author rebuttal***. This figure shows that HeO performs well across a wide range of step sizes $\gamma$, provided that the number of iterations $T$ is sufficient. This indicates that HeO is relatively insensitive to the step size $\gamma$.
>
>
> > 2. The paper demonstrates improved performance over several classic solvers like MDGE, SA, and LQA. However, it does not provide enough evidence or discuss the potential of HeO to achieve state-of-the-art performance on the studied CO problems
>
> Our goal is not to demonstrate that HeO achieves state-of-the-art performance for specific combinatorial optimization tasks, as metaheuristics can be tailored to achieve state-of-the-art results for particular problems [4]. Instead, we aim to highlight the generality of HeO: *it employs a different theoretical approach while performing competitively across a wide range of combinatorial optimization problems without the need for specialized design for each case*. This makes HeO a promising framework for addressing a broad spectrum of combinatorial optimization challenges.
>
> To achieve this, we have compared our HeO framework against various types of combinatorial optimization problems, including max-cut, 3-SAT, ternary network training, variable selection, and minimum vertex cover. We evaluated HeO alongside advanced methods proposed in recent years, including state-of-the-art higher-order Ising machines [5] and coherent Ising machines (CIMs) and their variants, which are widely used in industry for various combinatorial optimizations [6].
>
>
>
> > 3. The submission seems to be not incomplete as there lack of appendices or supplementary materials that verify the soundness of the theorems presented in Section 3.
>
>
> Thank you for pointing this out. We will include supplementary materials with detailed proofs for Theorems 1–3 and additional implementation details. Since revisions to the paper are not permitted during the rebuttal phase, we provide a brief outline of the proof in the *global author rebuttal*.
>
> [1] Liao F Y, Ding L, Zheng Y. Error bounds, PL condition, and quadratic growth for weakly convex functions, and linear convergences of proximal point methods[C]//6th Annual Learning for Dynamics & Control Conference. PMLR, 2024: 993-1005.
>
> [2] Evans L C. Partial differential equations[M]. American Mathematical Society, 2022.
>
> [3] Atenas F, Sagastizábal C, Silva P J S, et al. A unified analysis of descent sequences in weakly convex optimization, including convergence rates for bundle methods[J]. SIAM Journal on Optimization, 2023, 33(1): 89-115.
>
> [4] Bybee C, Kleyko D, Nikonov D E, et al. Efficient optimization with higher-order Ising machines[J]. Nature Communications, 2023, 14(1): 6033
>
> [5] Wang J, Ebler D, Wong K Y M, et al. Bifurcation behaviors shape how continuous physical dynamics solves discrete Ising optimization[J]. Nature Communications, 2023, 14(1): 2510.
>
> [6] Glover, Fred W., and Gary A. Kochenberger, eds. Handbook of metaheuristics. Vol. 57. Springer Science & Business Media, 2003.

---

> > ### Comment · Reviewer_uVQ4 · 2024-08-09
> >
> > Thank you for the rebuttal, which addresses most of my concerns. I agree that metaheuristics can be tailored to achieve state-of-the-art results for specific problems. Regarding this, could you please elaborate more on how the proposed HeO can be customized for a particular problem to enhance performance? This discussion could provide valuable insights for future work on extending the proposed solver framework.

---

> ### Author Response · Authors · 2024-08-10
>
> Thank you for your constructive comments. There is considerable flexibility and feasibility in customizing HeO for specific problems.
>
> First, as HeO is a gradient-based optimizer, it can be tailored to specific problems by designing more refined step schedules, such as adaptive step rules. Additionally, leveraging second-order optimization techniques like momentum (which we utilized in our paper) or Adam, as well as ensemble methods, can improve the overall performance of HeO on particular problem instances.
>
> Second, as discussed in Line 265, Section 5, we can customize HeO by designing a preconditioned matrix $A$ to reshape the heat equation. Prior knowledge about the problem can be embedded within the structure of $A$, such as by accounting for the relative importance between different dimensions of the discrete configuration $\mathbf{s}$ or by setting $A$ based on the Fisher information matrix of the parameter $\theta$. This approach can lead to a natural gradient descent method, enhancing the efficiency of the optimization process.
>
> Third, HeO allows for further customization by integrating problem-specific prior knowledge directly into the target function. By adding extra terms, we can improve the loss landscape or guide the search direction to meet particular purposes, thereby improving the quality of the solutions found.
>
> Fourth, HeO can be hybridized with other metaheuristic algorithms to explore the configuration space more effectively. Specifically, we can iteratively refine the solution by alternating between HeO and other metaheuristic update rules.
>
> We will incorporate these discussions into Section 5 in the revised version of the paper.

---

> > ### Comment · Reviewer_uVQ4 · 2024-08-13
> >
> > Thank you for providing the additional discussion. I am pleased to maintain my positive review.

---

### Official Review · Reviewer_c1oa · 2024-07-17

**Soundness:** 3
**Presentation:** 4
**Contribution:** 3
**Rating:** 8
**Confidence:** 4

**Summary:**

This paper aims to improve the efficiency of existing combinatorial optimization methods via heat diffusion. The author have made a thorough analysis over the existing problems and propose the heat diffusion method HOE for general combinatorial optimization problems. The empirical evaluation verifies its advantage over various combinatorial optimization problems.

**Strengths:**

1. The authors make a comprehensive analysis of the problems of existing methods, the proposed method is quite novel, with enough insights to the future research on the combinatorial optimization.
2. The proposed method is both theoretically supported and empirically justified.
3. The empirical evaluation spans a various of combinatorial optimization problems, and thorough analyses are presented.
4. The whole paper is well-written

**Weaknesses:**

1. In section 2, the analyses are only focused on the methods doing the gradient descent over the relaxed variables. It is unclear to me how the conclusions are generalized to the method like large neighborhood search, variable neighborhood search and path auxiliary sampling (as mentioned in the introduction part).

**Questions:**

See the weaknesses part

**Limitations:**

Yes

---

> ### Author Rebuttal · Authors · 2024-08-03
>
> > In section 2, the analyses are only focused on the methods doing the gradient descent over the relaxed variables. It is unclear to me how the conclusions are generalized to the method like large neighborhood search, variable neighborhood search and path auxiliary sampling.
>
> Good question. We acknowledge that generalizing the analysis from single-step search to multi-step search scenarios is challenging. In this paper, we focus on single-step search because, while multi-step search can improve the chances of escaping local minima and finding better solutions in general, it also increases computational costs, requires more careful design of search rules, and carries the risk of backtracking problems [1]. Integrating our HeO framework with multi-step search methods could be a valuable direction for future research, both theoretically and practically.
>
> [1] Sun H, Dai H, Xia W, et al. Path auxiliary proposal for MCMC in discrete space[C]//International Conference on Learning Representations. 2021.

---

> > ### Comment · Reviewer_c1oa · 2024-08-12
> >
> > I want to thank the authors for the response and will maintain my score.

---

### Official Review · Reviewer_NRJw · 2024-07-23

**Soundness:** 3
**Presentation:** 2
**Contribution:** 3
**Rating:** 6
**Confidence:** 5

**Summary:**

The paper presents a novel framework for solving combinatorial optimization problems using a concept termed "Heat diffusion optimization (HeO)." The approach diverges from traditional methods by utilizing heat diffusion to enhance information propagation within the solution space, allowing for more efficient problem-solving.

**Strengths:**

1.	The introduction of heat diffusion as a mechanism to aid in combinatorial optimization is novel and thoughtfully developed.
2.	Comprehensive experiments across different optimization problems illustrate the method's effectiveness and superiority over traditional approaches.
3.	The methodology is tested on a wide range of problems, showing its versatility and potential for broader application in real-world scenarios.

**Weaknesses:**

1.	The paper lacks a detailed discussion on the scalability of the method, especially in very large-dimensional spaces, which are common in real-world applications.
2.	More baselines are needed.

**Questions:**

1.	On Page 4, Line 129, the method for determining the value of K in the multilinear polynomial is not clear. Could you elaborate on how K is selected in different scenarios?
2.	Figure 2 shows that the energy does not monotonically decrease during optimization. Why the energy in Fig. 2 is not monotonically decreased?
3.	What are the algorithmic complexity and computational time of the proposed heat diffusion optimization method?
4.	Is it possible to apply your heat diffusion framework to other types of combinatorial optimization problems, such as those found in operational research?

**Limitations:**

1.	On Page 3, line 105, the definitions of \Delta_\theta and the function u are not given.
2.	The ability of global search is not theoretically analyzed.

---

> ### Author Rebuttal · Authors · 2024-08-03
>
> > 1. Could you elaborate on how K (Page 4, Line 129) is selected in different scenarios?
>
>
> Thank you for your valuable feedback. We will clarify this point after revision. The value of K is directly determined by the target function $f(\mathbf{s})$ to be optimized. For example:
>
> For minimum vertex cover problem, where $f(\mathbf{s})$ is a linear function (Eq. 18), we have K=1.
>
> For Quadratic unconstrained binary optimization, where $f(\mathbf{s})$ is a quadratic function (Eq. 14), we have K=2.
>
> For 3-satisfiability problem, where $f(\mathbf{s})$ is a cubic function (Eq. 15), we have K=3.
>
> > 2. Why does the energy in Fig. 2 not monotonically decrease?
>
> In simulated annealing, non-monotonicity arises due to the stochastic nature of the annealing process, which occasionally permits transitions from lower to higher energy states.
>
> In MCGE and our proposed HeO method, the non-monotonic behavior is a result of the stochasticity of the gradient estimate (Eq. 5 and Line 4-5 of Alg. 1 in the main paper).
>
> > 3. The algorithmic complexity, computational time, and the scalability of the HeO, especially in very large-dimensional spaces.
>
> Thank you for pointing this out. We will include a discussion about the computational complexity in the main paper. The complexity of our algorithm is relatively low. At each step, the most computationally intensive operation is calculating the gradient (Line 5 of Alg. 1 in the main paper). This can sometimes be expressed explicitly or be efficiently computed using automatic differentiation tools like PyTorch's autograd. The overall computational time is primarily dependent on the number of iterations $T$. As demonstrated in Fig. 1 of the ***PDF file of the global author rebuttal***, the time cost per iteration of our methods increases linearly with the problem dimension, with a small constant coefficient. This confirms the efficiency of our method. We will include this result in the Supplementary Information after revision.
>
> > 4. Is it possible to apply your heat diffusion framework to other types of combinatorial optimization problems?
>
> Yes, our HeO framework is versatile and can be applied to various types of combinatorial optimization problems. Since QUBO, 3-SAT, and Minimum Vertex Cover are NP-complete problems, any NP-hard combinatorial optimization problem can theoretically be encoded into these forms and then solved using HeO. For instance, our framework can be used to solve problems like the multidimensional knapsack problem and graph coloring.
>
> > 5. More baselines are needed.
>
> We have included representative baselines across various combinatorial optimization problems, including satisfiability (e.g., 3-SAT), graph theory (e.g., max-cut, minimum vertex cover), neural network training (e.g., ternary network training), and statistics (e.g., variable selection for linear regression). In our comparisons, the HeO framework performed better against advanced methods, including state-of-the-art higher-order Ising machines proposed last year [2], as well as coherent Ising machines (CIMs) and its variants, which are widely used in industry [3]. Our goal is not to claim that HeO surpasses specialized methods for specific combinatorial optimization tasks, as metaheuristics can be tailored to achieve state-of-the-art results for particular problems [4]. Instead, we aim to highlight the generality of HeO: *it offers a different theoretical approach while performing competitively across a wide range of combinatorial optimization problems without requiring specialized design* for each case. This makes HeO a promising framework for addressing a broad spectrum of combinatorial optimization challenges.
>
> > 6. On Page 3, line 105, the definitions of $\Delta_\theta$ and the function $u$ are not given.
>
> Thank you for pointing this out. We will clarify this after revision. The function $u$ is the solution to the heat equation as defined in Eq. (6), and $\Delta$ is the Laplace operator $\Delta_{\mathbf{x}} f(\mathbf{x}) = \sum_{i=1}^{n} \frac{\partial^2 f}{\partial x_i^2}$.
>
> > 7. The ability of global search is not theoretically analyzed.
>
> Thank you for pointing this out. We will include a theoretical discussion after revision. In general, finding the global minimum is not theoretically guaranteed for non-convex optimization problems [5], such as the combinatorial optimization problems studied in this paper. However, it can be demonstrated that the gradient of the target function under heat diffusion satisfies the inequality [6]:
>
> \begin{align}
> \left\| \bigtriangledown_{{\theta}}u(\tau,{\theta} )\right\| \leq \frac{C}{\sqrt{\tau}},
> \end{align}
>
> where the constant $C$ depends on the dimension. This implies that the target function becomes weakly convex, enabling the finding of global minima and faster convergence under certain conditions [7].
>
> [1] Korte B H, Vygen J, Korte B, et al. Combinatorial optimization[M]. Berlin: Springer, 2011.
>
> [2] Bybee C, Kleyko D, Nikonov D E, et al. Efficient optimization with higher-order Ising machines[J]. Nature Communications, 2023, 14(1): 6033.
>
> [3] Wang J, Ebler D, Wong K Y M, et al. Bifurcation behaviors shape how continuous physical dynamics solves discrete Ising optimization[J]. Nature Communications, 2023, 14(1): 2510.
>
> [4] Glover, Fred W., and Gary A. Kochenberger, eds. Handbook of metaheuristics. Vol. 57. Springer Science & Business Media, 2003.
>
> [5] Liao F Y, Ding L, Zheng Y. Error bounds, PL condition, and quadratic growth for weakly convex functions, and linear convergences of proximal point methods[C]//6th Annual Learning for Dynamics & Control Conference. PMLR, 2024: 993-1005.
>
> [6] Evans L C. Partial differential equations[M]. American Mathematical Society, 2022.
>
> [7] Atenas F, Sagastizábal C, Silva P J S, et al. A unified analysis of descent sequences in weakly convex optimization, including convergence rates for bundle methods[J]. SIAM Journal on Optimization, 2023, 33(1): 89-115.

---

### Author Rebuttal · Authors · 2024-08-03

We sincerely thank all the reviewers for their valuable feedback and suggestions. In addition to responding to each reviewer individually, we have included two figures in the ***PDF file*** of this global author rebuttal. Furthermore, to address concerns about theoretical soundness raised by Reviewer uVQ4, we provide a sketch of the proofs of the main theorems presented in Section 3 below.

Since revisions to the paper are not permitted during the rebuttal phase, we will include both the figures and the detailed proofs of the main theorems in the Supplementary Information after the revision.


###  Sketch of the Proof for Theorem 1

We need to show that for any $\tau > 0$, $u(0, \theta)=h(\theta)$ and $u(\tau, \theta)$ have the same global minimas.
It is straightforward to show that the global minima of $u(0, \theta)$ is also a global minima of $u(\tau, \theta)$ for any $\tau > 0$.
The proof for the converse direction relies on the backward uniqueness of the heat equation [1], which asserts that the initial state of a heat equation can be uniquely determined by its state at a time point $\tau$, provided some mild growth conditions, which are satisfied in our paper.
Denote $u(\tau,\mathbf{x};{\theta})=\mathbb{E}_{p(\mathbf{z})}[f(\mathrm{sgn}({\theta}-(\mathbf{x}+\sqrt{2\tau}\mathbf{z})))]$, where $p(\mathbf{z})$ obeys standard Gaussian distribution. If $\theta^\ast$ is one of the minimas of $h(\theta)$, and $\hat{\theta}$ is one of the minimas of $u(\tau,\theta)$, it can be proved that
\begin{align*}
u(\tau,\mathbf{x};\hat{{\theta}})=u(\tau,\mathbf{x};{{\theta}}^{\ast}),\quad \mathbf{x}\in \mathbb{R}^n
\end{align*}
Using the backward uniqueness of the heat equation, we have
\begin{align*}
    u(0,\mathbf{x};\hat{{\theta}})= u(0,\mathbf{x};{{\theta}}^{\ast}),\quad \mathbf{x}\in\mathbb{R}^n,
\end{align*}
that is
\begin{align*}
    h(\hat{{\theta}}) = h({{\theta}}^{\ast}).
\end{align*}
As a result, $\hat{{\theta}}$ is the one of minimas of $h({\theta})$.


###  Sketch of the Proof for Theorem 2

We first write the $u$ as

\begin{align*}
    u(\tau,\theta)
= \mathbb{E}_{p(\mathbf{x},\mathbf{z})} [f(\mathrm{sgn}(\theta+\sqrt{2\tau}\mathbf{z}-\mathbf{x}))],
\end{align*}

where $\mathbf{z} \sim N(\mathbf{0},I)$ is independent of $\mathbf{x}$. Using the property of multi-dimensional Gaussian integral [2], we have

\begin{align*}
    \mathbb{E}_{p(\mathbf{z})}[f(\mathrm{sgn}({\theta}+\sqrt{2\tau}\mathbf{z}-\mathbf{x}))]=f(\tilde{\mathbf{s}}),
\end{align*}

where $\tilde{\mathbf{s}}$ is a random vector determined by $\mathbf{x}$

\begin{align*}
    \tilde{s}_i = \mathrm{erf}(\frac{\theta_i - x_i}{\sqrt{2\tau}}),
\end{align*}

and $\mathrm{erf}(\cdot)$ is the error function. Therefore, we have

\begin{align*}
    u(\tau,{\theta}) = \mathbb{E}_{p(\mathbf{x})}[f(\mathrm{erf}(\frac{{\theta} - \mathbf{x}}{\sqrt{2\tau}}))]
\end{align*}

where $\mathrm{erf}(\cdot)$  is the element-wise error function.

###  Sketch of the Proof for Theorem 3

Define the square loss of ${\theta}$ as $e({\theta}) = (h({\theta})-h({\theta}^{\ast}))^2$
and the error function
\begin{align*}
    r(\tau,\mathbf{x};{\theta})  =  u(\tau,\mathbf{x};{\theta}) - u(\tau,\mathbf{x};{\theta}^{\ast})
\end{align*}

Define the energy function of the error function $r(\tau,\mathbf{x};{\theta})$ as
\begin{align*}
    E(\tau;{\theta}) = \int_{\mathbb{R}^n}r^2(\tau,\mathbf{x};{\theta})p(\mathbf{x})d\mathbf{x}.
\end{align*}

Use the Harnack's inequality[3] and integration by parts, we have for $0<\tau_1<\tau_2$
\begin{align*}
  E(\tau_1;{\theta}) \leq E(\tau_2;{\theta}) +  \frac{n}{2}\int_{\tau_1}^{\tau_2}\frac{ E(\tau;{\theta})}{\tau} d\tau.
\end{align*}
Using the Minkowski inequality on the measure $p(\mathbf{x})$, we have
\begin{align*}
      h({\theta})-h({\theta}^{\ast}) \leq
        \big(\int_{\mathbb{R}^n}  (f(\mathrm{sgn}({\theta}-\mathbf{x}))-u(\tau_1;\mathbf{x};{\theta}))^2 p(\mathbf{x})d\mathbf{x}\big)^{1/2}
     +
     \big(\int_{\mathbb{R}^n}  (f(\mathrm{sgn}({\theta}^{\ast}-\mathbf{x}))-u(\tau_1;\mathbf{x};{\theta}^{\ast}))^2 d\mathbf{x}\big)^{1/2}+E^{1/2}(\tau_1;{\theta}).
\end{align*}
Using the continuity of the heat operator, given $\epsilon>0$, there exists a $\tau_1>0$, such that
\begin{align*}
\begin{aligned}
    \big(\int_{\mathbb{R}^n}  (f(\mathrm{sgn}({\theta}-\mathbf{x}))-u(\tau_1;\mathbf{x};{\theta}))^2 p(\mathbf{x})d\mathbf{x}\big)^{1/2}
     +\big(\int_{\mathbb{R}^n}  (f(\mathrm{sgn}({\theta}^{\ast}-\mathbf{x}))-u(\tau_1;\mathbf{x};{\theta}^{\ast}))^2p(\mathbf{x}) d\mathbf{x}\big)^{1/2}<\epsilon.
\end{aligned}
\end{align*}
 We then have the error control for $e({\theta})$:
\begin{align*}
    e^{1/2}({\theta}) \leq E^{1/2}(\tau_1;{\theta}) + \epsilon \leq  \big(E(\tau_2;{\theta}) +\frac{n}{2}\int_{\tau_1}^{\tau_2}\frac{ E(\tau;{\theta})}{\tau} d\tau \big)^{1/2}+ \epsilon.
\end{align*}
Noticed that
\begin{align*}
    E(\tau;{\theta})\leq (\breve{f}-f^{\ast})(u(\tau,{\theta}^{\ast})-u(\tau,{\theta})),
\end{align*}
where $ \breve{f}=\max_{\mathbf{s}} f(\mathbf{s}),f^\ast=\min_{\mathbf{s}} f(\mathbf{s})$, and we prove the theorem.

[1] Jie Wu and Liqun Zhang. Backward uniqueness for general parabolic operators in the whole space. Calculus of Variations and Partial Differential Equations, 58:1–19, 2019.

[2] Mobahi H, Fisher J W. On the link between gaussian homotopy continuation and convex envelopes[C]//Energy Minimization Methods in Computer Vision and Pattern Recognition: 10th International Conference, EMMCVPR 2015, Hong Kong, China, January 13-16, 2015. Proceedings 10.

[3] Evans L C. Partial differential equations[M]. American Mathematical Society, 2022.

---

### Decision · Program_Chairs · 2024-09-25

**Decision:**

Accept (poster)

**Comment:**

This paper introduces a novel approach to enhancing combinatorial optimization through heat diffusion, effectively transforming the target function to facilitate better information flow back to the solver. The reviewers unanimously found the idea to be novel and promising. During the discussion, the authors provided clarification and additional theoretical analysis that addressed most concerns.